# GROUPED-HEAD LATENT ATTENTION

## ABSTRACT

Attention mechanisms underpin the success of large language models (LLMs), yet their substantial computational and memory overhead poses challenges for optimizing efficiency and performance. A critical bottleneck arises as KV cache and attention computations scale rapidly with text length, challenging deployment on hardware with limited computational and memory resources. We observe that attention mechanisms exhibit substantial redundancy, since the KV cache can be significantly compressed and attention maps across heads display high similarity, revealing that much of the computation and storage is unnecessary. Leveraging these insights, we propose **G**rouped-Head Laten**T A**ttention (GTA), a novel attention mechanism that reduces memory usage and computational complexity while maintaining performance. GTA comprises two components: (1) a shared attention map mechanism that reuses attention scores across multiple heads, decreasing the key cache size; and (2) a nonlinear value decoder with learned projections that compresses the value cache into a latent space, further cutting memory needs. GTA cuts attention computation FLOPs by up to *62.5%* versus Grouped-Query Attention and shrink the KV cache by up to *70%*, all while avoiding the extra overhead of Multi-Head Latent Attention to improve LLM deployment efficiency. Consequently, GTA models achieve a *2×* increase in end-to-end inference speed, with prefill benefiting from reduced computational cost and decoding benefiting from the smaller cache footprint.

## 1 INTRODUCTION

Large language models (LLMs) have revolutionized natural language processing, driving break-throughs in text generation, reasoning, and contextual understanding (Brown et al., 2020; Touvron et al., 2023). The attention mechanism, a core component of these models, enables selective focus on relevant parts of the input sequence, underpinning their expressive power (Vaswani et al., 2017). However, the memory and computational demands of attention, particularly the key-value (KV) cache in autoregressive generation, pose significant challenges for long-context scenarios and resource-constrained environments (Dao et al., 2022; Liu et al., 2023). These bottlenecks limit the scalability of LLMs in practical applications, where memory efficiency and low-latency inference are critical.

Prior efforts to mitigate attention-related challenges in large language models (LLMs) have led to several innovations. Multi-Head Attention (MHA) (Vaswani et al., 2017), the foundation of modern transformers, projects input sequences into multiple query, key, and value representations to capture diverse contextual patterns, but its KV cache scales poorly with sequence length, limiting long-context applicability. Multi-Query Attention (MQA) (Shazeer, 2019) reduces memory usage by sharing a single key-value pair across heads, yet sacrifices expressivity. Grouped-Query Attention (GQA) (Ainslie et al., 2023) groups heads to balance efficiency and performance, but compromises attention granularity. Multi-head Latent Attention (MLA) (Liu et al., 2024a) compresses the KV cache while preserving representational capacity, but its high computational overhead restricts use in resource-constrained settings. Other methods, such as differential attention (Ye et al., 2025) and convolution-augmented attention (Golovneva et al., 2025), improve contextual focus, but often increase complexity. These approaches are limited by high computational overhead, inefficient KV cache storage, and compromised model performance, with no method optimizing all three simultaneously.

To address this limitations, we propose **Grouped-head latenT Attention (GTA)**, a novel attention framework that optimizes memory usage and computational efficiency while preserving the expressive

power of MHA. GTA introduces two key innovations, as detailed in our method. First, it employs a shared attention map mechanism, grouping query and key projections to reuse computations across heads, thereby reducing computational overhead while maintaining fine-grained attention patterns. Second, it leverages a nonlinear value decoder that compresses the value cache into a compact latent space, using a context-adaptive sigmoid gate to dynamically generate head-specific values (Shazeer, 2020b). This design, illustrated in our architectural diagrams, significantly reduces memory requirements compared to traditional attention mechanisms, enabling efficient inference without sacrificing model quality. By combining grouped projections with nonlinear decoding, GTA achieves robust expressivity, overcoming the trade-offs observed in GQA and MLA.

In this paper, we show the design roadmap of GTA, and present experiments on GTA models ranging from 160M to 1B parameters. Not only the statistical validation of GTA's efficiency is provided the practical evaluations of cache footprint and latency are also carried out. The contributions of this work are as follows:

- Proposal of GTA, a novel attention mechanism that reduces self-attention computation by up to **62.5%** and KV cache size by up to **70%** while preserving expressive power through shared attention maps and nonlinear decoding.

- Training of GTA models on large-scale corpora and validation of their performance, matching or surpassing GQA on benchmarks across model scales from **160M to 1B** parameters.

- Analysis of GTA's inference speed in prefill and decode stages, demonstrating **2×** throughput compared to GQA, validating its effectiveness for low-latency LLM deployment. By breaking the conventional trade-off between efficiency and expressivity, GTA paves the way for scalable, sustainable, and high-performance LLM deployment in various devices.

- This paper record the attention mechanism design process, including detailed design introduction, analysis methods, and evaluation procedures, guiding future efficient attention designs.

## 2 RELATED WORK

Attention mechanisms are central to LLMs, enabling effective modeling of contextual dependencies (Vaswani et al., 2017). However, the KV cache in standard attention mechanisms scales linearly with sequence length, creating memory and computational bottlenecks (Dao et al., 2022). Recent research has developed dense attention variants to optimize KV cache usage through sharing or compression, aligning with GTA. We review these approaches, focusing on methods that share KV caches across heads or layers and those that use latent compression, positioning GTA's contributions.

**Shared KV cache methods.** Several methods reduce memory usage by sharing KV caches across heads or layers. MHA (Vaswani et al., 2017), the transformer baseline, uses independent KV caches for each head, resulting in high memory demands. MQA (Shazeer, 2019) shares a single KV pair across all heads, significantly reducing memory but limiting expressivity. GQA (Ainslie et al., 2023) groups heads and shares KV pairs within each group, balancing efficiency and performance, as seen in LLaMA (Touvron et al., 2023). You Only Cache Once (YOCO) (Sun et al., 2024) employs a decoder-decoder architecture to cache KV pairs once, sharing them across layers via cross-attention, reducing memory while maintaining global attention. These methods trade off some expressivity for efficiency, which GTA addresses through its design.

**Latent attention mechanisms.** Another approach compresses the KV cache using latent representations. MLA used in DeepSeek-V3 (Liu et al., 2024a) and PLM (Deng et al., 2025), compresses keys and values into a latent vector, achieving significant memory savings while preserving performance. Similarly, GTA uses a compressed latent value representation with a nonlinear decoder to generate head-specific values, enhancing expressivity with low computational costs. GTA's nonlinear decoding, inspired by gated mechanisms like GLU (Shazeer, 2020a) and GLA (Yang et al., 2024), distinguishes it by maximizing information density.

**Performance-focused attention.** Some methods prioritize performance over efficiency. Multi-Token Attention (MTA) (Golovneva et al., 2025) uses convolutions to enhance contextual interactions, and the Differential Transformer (Ye et al., 2025) employs dual softmax maps for sharper focus. These approaches improve accuracy but often increase computational complexity, unlike GTA's efficiency-driven design.

**Comparison with Zadouri et al. (2025)** The paper (Zadouri et al., 2025) introduces Grouped Tied Attention, which reduces cache requirements by sharing key and value components, thereby increasing arithmetic intensity. Building on this, Grouped Latent Attention is proposed to enhance model parallelism through grouped operations on latent variables within the MLA framework. In contrast, **G**rouped-Head Laten**T A**ttention (GTA) proposed in this paper adopts a novel attention matrix sharing strategy combined with a nonlinear value decoding process. To our knowledge, this is the first approach to achieve simultaneous improvements in both the prefill and decode phases without compromising model quality.

## 3 METHOD

In this section, we present our proposed **Grouped-Head Latent Attention (GTA)** mechanism, which enhances the efficiency of transformer architectures while retaining their expressive power. We begin by revisiting Multi-Head Attention (MHA) and introducing our efficiency-driven variants, Grouped-Value Attention (GVA) and Grouped-Head Attention (GHA). These approaches progressively reduce memory and computational overheads but introduce trade-offs in expressivity. Building on their insights, we introduce GTA, which employs a compressed latent representation and a nonlinear decoder to achieve superior efficiency and performance.

### 3.1 EVOLVING PATTERNS OF ATTENTION MECHANISMS

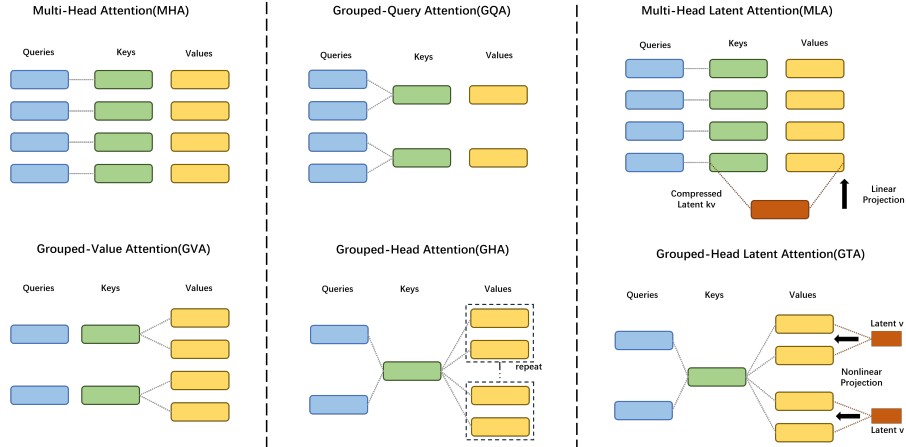

Figure 1: Attention Architecture: Comparing MHA with GVA and GHA, highlighting key, query, and value projection differences. Left-to-right: cache reduction via sharing and compression; top-to-bottom: attention computation reduction via shared attention maps and non-linearity.

**Brief introduction to MHA** MHA (Vaswani et al., 2017) underpins modern transformers by enabling the model to attend to diverse sequence patterns. For an input $X \in \mathbb{R}^{N \times H}$, where $N$ denotes sequence length and $H$ the hidden dimension, MHA projects $X$ into queries, keys, and values:

$$Q = XW_Q \in \mathbb{R}^{N \times n_h d_h}, K = XW_K \in \mathbb{R}^{N \times n_h d_h}, V = XW_V \in \mathbb{R}^{N \times n_h d_h}, \tag{1}$$

where $W_Q, W_K, W_V \in \mathbb{R}^{H \times n_h d_h}$ are projection matrices, $n_h$ is the number of heads, and $d_h$ satisfies $n_h \cdot d_h = H$. Each head computes:

$$O_i = \text{Softmax}\left(\frac{Q_i K_i^T}{\sqrt{d_h}}\right) V_i W_{O_i} \in \mathbb{R}^{N \times H}, \tag{2}$$

with $W_{O_i} \in \mathbb{R}^{d_h \times H}$ as the output projection, yielding $O = \sum_{i=1}^{n_h} O_i$. While effective, MHA's key-value (KV) cache grows as $\mathcal{O}(2HN)$, posing scalability challenges for long sequences.

To address these inefficiencies, techniques such as Multi-Query Attention (MQA) (Shazeer, 2019) and Grouped-Query Attention (GQA) (Ainslie et al., 2023) emerged, reducing memory overhead by sharing keys and values across heads. Building on this foundation, we introduce GVA and GHA as evolutionary steps toward our novel GTA mechanism.

**Grouping Values to Share Attention Matrix** In GVA, the attention weights computed from queries and keys are shared across groups of heads. This means that multiple heads within a group apply the same attention distribution but operate on distinct value projections. By reusing the attention weights, GVA reduces redundant computation while preserving the ability of each head to produce unique outputs through its own value transformation. This strikes a balance between efficiency and representational flexibility, though it still requires maintaining a full set of value projections, keeping memory usage relatively high.

**Grouping Heads to Compress Attention** GHA extends this idea by sharing query and key representations within groups of heads, while deriving distinct value representations for each head. Specifically, multiple heads in a group use the same query and key representations, but their values are computed separately from a shared source, further compressing the memory footprint of the KV cache. This sharing mechanism significantly lowers both computational and storage costs, making GHA well-suited for resource-constrained settings. However, the reduced diversity in query and key representations can limit the model's ability to capture fine-grained dependencies, potentially impacting performance on complex tasks.

The progression from MHA to GVA and GHA illustrates a critical trade-off between efficiency and expressivity in attention mechanisms. These insights motivate the development of GTA, which introduces a novel nonlinear decoder to achieve greater efficiency without sacrificing performance, addressing the limitations of its predecessors.

## 3.2 GROUPED-HEAD LATENT ATTENTION

GHA mitigates the computational and memory demands of MHA by sharing query, key, and value representations across heads, but this often compromises expressivity due to fewer unique representations. To address this limitation, we propose GTA, a novel mechanism that enhances efficiency while preserving representational power. By integrating a compressed latent value representation with a nonlinear decoder, GTA dynamically generates head-specific values, achieving robust expressivity with a reduced memory footprint. This design makes GTA particularly suited for resource-constrained inference.

**Input projections and grouping** GTA begins by processing an input sequence $X \in \mathbb{R}^{N \times H}$, where $N$ is the sequence length and $H$ is the hidden dimension. It computes queries, keys, and a compressed latent value representation as follows:

$$Q = XW_Q \in \mathbb{R}^{N \times n_q d_h}, \quad K = XW_K \in \mathbb{R}^{N \times n_k d_h}, \quad C = XW_C \in \mathbb{R}^{N \times n_c d_l}, \quad (3)$$

where $W_Q \in \mathbb{R}^{H \times n_q d_h}$, $W_K \in \mathbb{R}^{H \times n_k d_h}$, and $W_C \in \mathbb{R}^{H \times n_c d_l}$ are projection matrices. Here, $n_q$, $n_k$, and $n_c$ represent the number of query, key, and value groups, while $d_h$ and $d_l$ denote the head and latent dimensions, with $d_l \geq d_h$ to ensure expressive projections.

To enhance efficiency, GTA organizes these representations into groups. Queries are divided into $n_q$ groups, with each head $i$ using $Q_{q(i)} \in \mathbb{R}^{N \times d_h}$ via a mapping $q(i)$. Keys are partitioned into $n_k$ groups, with head $i$ accessing $K_{k(i)} \in \mathbb{R}^{N \times d_h}$ via a mapping $k(i)$. Values are derived from the latent representation $C$, split into $n_c$ groups, with head $i$ using $C_{c(i)} \in \mathbb{R}^{N \times d_l}$ from group $c(i)$. This hierarchical grouping minimizes redundancy, preserves flexible attention patterns, and paves the way for efficient value generation.

**Nonlinear value decoder** Building on this grouped structure, GTA generates head-specific value matrices $V_i \in \mathbb{R}^{N \times d_h}$ for each head $i$:

$$V_i = C_{c(i)} W_{P,i} \odot \text{Sigmoid}(x_t W_{G,i}), \quad (4)$$

where $W_{P,i} \in \mathbb{R}^{d_l \times d_h}$ is a head-specific projection matrix, $W_{G,i} \in \mathbb{R}^{H \times d_h}$ is a gating matrix, and $x_t \in \mathbb{R}^H$ is the current token's representation.

The gate $\text{Sigmoid}(x_t W_{G,i}) \in \mathbb{R}^{d_h}$, broadcasting across the sequence, introduces nonlinearity through element-wise multiplication ($\odot$). For each head $i$, GTA generates the value $V_i \in \mathbb{R}^{N \times d_h}$ from the compressed latent representation $C_{c(i)} \in \mathbb{R}^{N \times d_l}$, where $c(i)$ assigns head $i$ to one of $n_c$ value groups.

The projection is performed using $W_{P,i} \in \mathbb{R}^{d_l \times d_h}$, which combines a direct mapping for a subset of $C_{c(i)}$'s elements—determined by the head and group assignment—with a learnable component initialized with small random values to enhance diversity. The resulting projection, $C_{c(i)}W_{P,i}$, is then modulated by the gate, introducing nonlinearity and enabling context-adaptive feature selection. This design ensures full-rank projections, preventing information loss and enhancing the diversity of the final output across heads within the same group. The nonlinear decoding process thus enables GTA to produce expressive, context-sensitive values for attention computation.

## 3.3 EFFICIENT ATTENTION COMPUTATION

Using the dynamically generated values, GTA computes the attention output for each head $i$:

$$O_i = \text{Softmax}\left(\frac{Q_i K_{k(i)}^T}{\sqrt{d_h}}\right) V_i W_{O,i}, \tag{5}$$

where $W_{O,i} \in \mathbb{R}^{d_h \times H}$ is the output projection, and the final output is $O = \sum_{i=1}^{n_h} O_i$. For efficient inference, GTA reformulates the computation:

$$O_i = \left(\text{Softmax}\left(\frac{Q_i K_{k(i)}^T}{\sqrt{d_h}}\right) C_{c(i)} W_{P,i}\right) \odot \text{Sigmoid}(x_t W_{G,i}) W_{O,i}. \tag{6}$$

GTA caches both the compressed latent values $C \in \mathbb{R}^{N \times n_c d_l}$ and keys $K \in \mathbb{R}^{N \times n_k d_h}$, resulting in a memory footprint of $\mathcal{O}((n_c d_l + n_k d_h)N)$. This design reduces memory usage compared to traditional grouped attention mechanisms, while computing the nonlinear gate on-the-fly using $x_t$, thereby minimizing computational overhead. Furthermore, GTA's nonlinear decoder enhances expressivity over linear projections by combining a compact latent representation with a context-aware sigmoid gate, improving output diversity, akin to increasing the effective rank (Shazeer, 2020a). This architecture achieves a robust balance of scalability, expressivity, and efficiency, making GTA a compelling solution for resource-constrained tasks.

# 4 PERFORMANCE EVALUATION

To evaluate the effectiveness of our proposed GTA approach, we conduct extensive experiments on language model pretraining with varying model sizes and sequence lengths. We analyze performance in terms of evaluation loss, parameter count, and memory efficiency of KV cache. Additionally, we perform ablation studies to investigate the impact of specific design choices.

## 4.1 VALIDATING GTA EFFECTIVENESS

We train transformer language models on the C4 dataset (Raffel et al., 2023) using sequence lengths of 2048 and 4096 tokens. Training employs the AdamW optimizer (Loshchilov & Hutter, 2017) with cosine scheduler and the TinyLlama tokenizer (Zhang et al., 2024). Full training details are provided in Appendix A.1 and Appendix A.2. To benchmark our GTA, we compare it against the following attention variants: MHA (Vaswani et al., 2017), GQA (Ainslie et al., 2023) and MLA (Liu et al., 2024a).

Prior work often adjusts model parameters (e.g., hidden state dimensions) to match total parameter counts across architectures, but this can confound the analysis of attention mechanisms by altering MLP capacity. To isolate the impact of attention, we adopt a framework that fixes non-attention parameters (e.g., hidden state dimensions, MLP sizes) across models, allowing parameter count variations solely due to attention design. This ensures a controlled comparison, focusing on the attention mechanism's contribution to performance and efficiency.

**Results for 160M parameter models.** Table 1 presents the performance of models with approximately 160M parameters. At a sequence length of 2048 tokens, GTA (with the GTA2 configuration) achieves a lower evaluation loss and better Wikitext perplexity (PPL) compared to MHA, GQA, and MLA. Additionally, GTA (with the GTA1 configuration) records higher downstream task accuracy, demonstrating a notable improvement. These results are achieved using only 12.5% of MHA's KV cache size per layer (192 vs. 1536 dimensions), highlighting GTA's memory efficiency. At a

sequence length of 4096 tokens, GTA remains competitive, delivering the lowest evaluation loss and comparable PPL, alongside the highest average downstream accuracy. This indicates GTA's ability to maintain strong performance with reduced memory requirements for longer sequences.

Table 1: Performance of 160M parameter models at sequence lengths of 2048 and 4096. This table compares models based on total parameter count, KV cache dimensions per layer, evaluation loss, and average accuracy across a suite of downstream tasks.

| Model | Params | Cache/layer | Seq Len | Eval Loss | Wikitext PPL | PIQA | HellaSwag | ARC-e | ARC-c | Winogrande | Avg |
|-------|--------|-------------|---------|-----------|--------------|------|-----------|-------|-------|------------|-----|
| GQA | 158.50M | $384 (3 \times 2 \times 64)$ | 2048 | 2.719 | 23.63 | 65.94 | 30.70 | 42.59 | 19.53 | 51.38 | 42.03 |
| MLA | 172.54M | 288 (256+32) | 2048 | 2.707 | 22.69 | 65.01 | 30.72 | 40.65 | 19.19 | 51.38 | 41.39 |
| MHA | 178.78M | $1536 (12 \times 2 \times 64)$ | 2048 | 2.696 | 23.03 | 66.26 | 30.87 | 42.85 | 19.49 | 52.17 | 42.33 |
| GTA1 | 160.75M | 192 (64+128) | 2048 | 2.712 | 22.67 | 65.72 | 31.42 | 41.58 | 19.45 | 53.59 | **42.41** |
| GTA2 | 164.13M | 192 (64+128) | 2048 | **2.690** | **22.41** | 65.72 | 31.42 | 41.58 | 19.45 | 53.59 | 42.35 |
| GQA | 158.50M | $384 (3 \times 2 \times 64)$ | 4096 | 2.831 | 26.93 | 63.71 | 29.28 | 39.27 | 18.26 | 49.96 | 40.09 |
| MLA | 172.54M | 288 (256+32) | 4096 | 2.823 | 24.98 | 64.09 | 29.52 | 38.89 | 18.43 | 50.75 | 40.33 |
| MHA | 178.78M | $1536 (12 \times 2 \times 64)$ | 4096 | 2.827 | 25.16 | 63.87 | 29.38 | 39.56 | 18.77 | 49.67 | 40.25 |
| GTA1 | 160.75M | 192 (64+128) | 4096 | 2.819 | **24.01** | 63.82 | 29.53 | 39.48 | 18.60 | 52.80 | **40.85** |
| GTA2 | 164.13M | 192 (64+128) | 4096 | **2.812** | 25.06 | 63.71 | 29.30 | 38.85 | 20.48 | 51.30 | 40.73 |

**Results for 500M parameter models.** Table 2 summarizes results for models with approximately 500M parameters. At 2048 tokens, GTA achieves a lower evaluation loss and higher downstream accuracy, with competitive PPL relative to MHA and GQA. This performance is attained with only 12.5% of MHA's KV cache size (320 vs. 2560 dimensions). Configurations with smaller caches (e.g., 192 dimensions, 7.5% of MHA's) yield comparable results, balancing performance and efficiency. At 4096 tokens, GTA not only matches MHA's evaluation loss but also provides lower Wikitext PPL and higher downstream accuracy. Its reduced memory footprint remains a key benefit.

Table 2: Performance of 500M parameter models at sequence lengths of 2048 and 4096. This table compares models based on total parameter count, KV cache dimensions per layer, evaluation loss, and average accuracy across a suite of downstream tasks.

| Model | Params | Cache/layer | Seq Len | Eval Loss | Wikitext PPL | PIQA | HellaSwag | ARC-e | ARC-c | Winogrande | Avg |
|-------|--------|-------------|---------|-----------|--------------|------|-----------|-------|-------|------------|-----|
| GQA | 483.23M | $512 (4 \times 2 \times 64)$ | 2048 | 2.508 | 18.52 | 68.61 | 34.31 | 46.72 | 20.44 | 51.62 | 44.34 |
| MLA | 516.00M | 342 (320+32) | 2048 | 2.486 | **16.44** | 68.77 | 34.52 | 45.86 | 19.45 | 53.43 | 44.41 |
| MHA | 543.27M | $2560 (20 \times 2 \times 64)$ | 2048 | 2.484 | 17.53 | 68.44 | 35.11 | 47.35 | 20.73 | 50.91 | 44.51 |
| GTA3 | 486.98M | 192 (64+128) | 2048 | 2.503 | 17.34 | 68.50 | 34.22 | 46.84 | 19.80 | 50.28 | 43.92 |
| GTA4 | 500.11M | 320 (64+256) | 2048 | **2.478** | 16.82 | 68.55 | 34.93 | 47.05 | 20.99 | 53.51 | **45.01** |
| GQA | 483.23M | $512 (4 \times 2 \times 64)$ | 4096 | 2.614 | 19.01 | 67.41 | 31.97 | 43.86 | 18.43 | 52.17 | 42.77 |
| MLA | 516.00M | 342 (320+32) | 4096 | 2.596 | 17.99 | 65.78 | 32.29 | 44.28 | 19.20 | 52.88 | 42.89 |
| MHA | 543.27M | $2560 (20 \times 2 \times 64)$ | 4096 | **2.592** | 19.87 | 66.65 | 32.79 | 43.98 | 19.37 | 51.62 | 42.88 |
| GTA3 | 486.98M | 192 (64+128) | 4096 | 2.609 | 18.77 | 67.25 | 31.85 | 44.49 | 18.26 | 51.07 | 42.58 |
| GTA4 | 500.11M | 320 (64+256) | 4096 | **2.592** | **16.96** | 66.97 | 32.45 | 43.94 | 18.26 | 53.18 | **42.96** |

## 4.2 SCALING TO 1B LANGUAGE MODEL

To investigate the impact of scaling model size and training data, we train two models, GTA-1B and GQA-1B, each with 1 billion parameters, trained on 220 billion tokens from the smollm-corpus (Ben Allal et al., 2024) dataset, with details in Appendix A.1. GQA-1B adopts the LLaMA-3.2 (llama team, 2024) framework with MobileLLM's (Liu et al., 2024b) optimal hyperparameters, tuned via extensive search. GTA-1B, designed for efficiency, uses only 30% of GQA-1B's cache size while maintaining competitive performance.

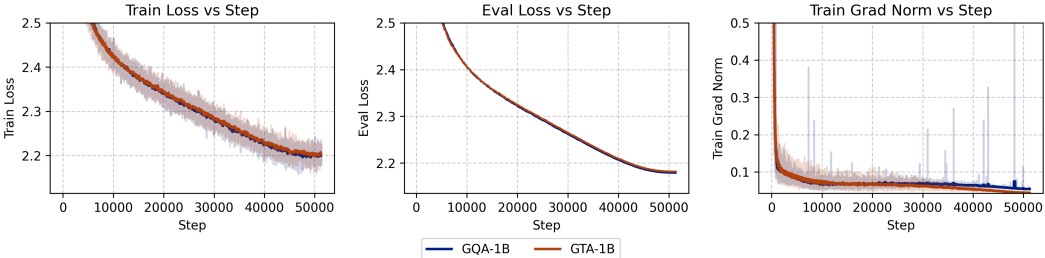

Figure 2: Loss and gradient norm curves over 50,000 training steps for GTA-1B and GQA-1B, showing stable convergence with GTA-1B's reduced cache size.

Figure 2 shows the training curves, with both models converging stably. GTA-1B's loss trajectory matches GQA-1B's, despite its reduced cache, highlighting its memory-efficient architecture. We leverage lm-evaluation-harness (Gao et al., 2024) to evaluate our models. These evaluation

datasets can be divide into: general tasks (ARC-e, ARC-c (Clark et al., 2018), HellaSwag (Zellers et al., 2019), BoolQ (Clark et al., 2019), PIQA (Bisk et al., 2020), MathQA (Amini et al., 2019), TruthfulQA (Lin et al., 2021), SIQA (Sap et al., 2019)); coding task (MBPP (Austin et al., 2021)); instruction following task (IFEval (Zhou et al., 2023)); reasoning tasks (LogiQA (Liu et al., 2020), BBH (Suzgun et al., 2022));

Table 3: We evaluate our models with several common and domain benchmarks, the vertical line denotes different few-shot numbers, where the left ones use 5-shot and the right ones use 3-shot.

| Model | PIQA | HellaS. | LogiQA | SIQA | ARC-e | ARC-c | BoolQ | MathQA | TQA | BBH | IFEval | MBPP | Avg. |
|---|---|---|---|---|---|---|---|---|---|---|---|---|---|
| GQA-1B | 75.03 | 46.46 | 24.42 | 46.26 | 77.02 | 42.58 | 63.89 | 25.56 | 40.48 | 23.01 | 9.90 | 12.80 | **40.62** |
| GTA-1B | 74.59 | 46.47 | 23.50 | 44.26 | 75.63 | 40.87 | 62.01 | 25.93 | 39.01 | 21.01 | 9.80 | 11.60 | 39.56 |
| GQA-1B-SFT | 74.31 | 45.52 | 20.58 | 42.42 | 70.45 | 36.09 | 63.57 | 26.26 | 40.89 | 22.01 | 29.76 | 15.80 | 40.64 |
| GTA-1B-SFT | 74.59 | 45.20 | 19.80 | 45.08 | 71.30 | 39.16 | 65.01 | 26.47 | 41.30 | 25.50 | 36.04 | 16.60 | **42.17** |

For supervised fine-tuning (SFT), we further train both base models using the tulu3 dataset (Lambert et al., 2024), a diverse collection of instruction-tuning data designed to enhance model generalization across tasks. The fine-tuned models, GTA-1B-SFT and GQA-1B-SFT, are evaluated on the same benchmarks. Table 3 shows that GTA-1B-SFT delivers performance comparable to GQA-1B-SFT across diverse benchmarks, with a notable improvement in average accuracy. This competitive performance, combined with GTA-1B's reduced cache size, highlights its ability to generalize effectively during fine-tuning under resource constraints.

In summary, GTA-1B achieves comparable performance to GQA-1B in both base and fine-tuned settings, using only 30% of GQA-1B's KV cache size and 37.5% of its self-attention computational cost. These results underscore the potential of memory- and compute-efficient architectures for scaling large language models, enabling sustainable and resource-efficient AI development.

### 4.3 ABLATION STUDIES ON GTA COMPONENTS

We perform ablation studies to evaluate the sensitivity of our GTA to critical parameters: attention matrix sharing, head dimension, and nonlinearity choice. We systematically analyze three key components: Shared Attention (SA), Nonlinear decoding (NL), and Up-projection (UP).

Figure 3 illustrates the trade-off between average performance and total latency for various GTA configurations on 500M parameter models with 2048 sequence length. From the plot, we observe that the nonlinear decoding (NL) has a significant impact on model performance, leading to substantial gains. The shared attention mechanism (SA) greatly affects speed by reducing latency. Meanwhile, the up-projection (UP) improves model performance with minimal increase in latency. The full GTA configuration achieves strong performance while optimizing the latency-performance balance. Key findings include: (1) sharing attention matrices across heads reduces parameters and slightly improves performance when combined with other components, suggesting a regularization benefit; (2)

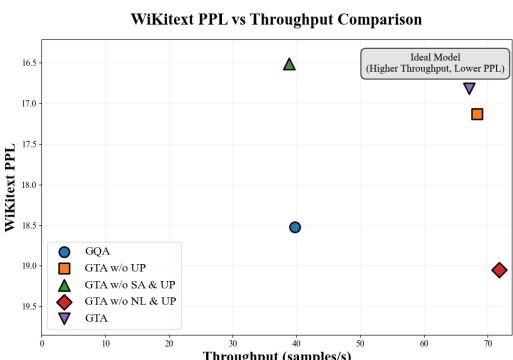

Figure 3: Performance vs. Total Latency Comparison for GTA components.

increasing head dimension enhances performance for both GTA and GQA, with GTA consistently outperforming GQA; and (3) Sigmoid nonlinearity outperforms sparser alternatives (e.g., SiLU, ReLU²), emphasizing the need for higher-rank value representations. Comprehensive results and configurations are detailed in Appendix A.4. **GTA demonstrates the ability to increase throughput while preserving modeling capability and overall performance.**

## 5 EFFICIENCY EVALUATION

In this section, we evaluate the computational and memory efficiency of our GTA mechanism against prominent attention variants: MHA, GQA, MLA, GVA, and GHA. Through theoretical analysis

and empirical benchmarks, we demonstrate GTA's ability to achieve high expressivity with reduced resource demands, positioning it as an efficient solution for modern transformer architectures.

## 5.1 THEORETICAL EFFICIENCY ANALYSIS

Table 4 compares GTA with existing attention mechanisms across memory usage, computational complexity, and expressivity, with detailed formulations in Appendix B. GTA achieves a favorable efficiency-expressivity trade-off: while GHA has the lowest overhead, it suffers from weak expressivity. In contrast, GTA maintains strong expressivity comparable to MHA while achieving substantial efficiency improvements.

Table 4: Efficiency comparison of attention mechanisms. Lower numbers indicate better efficiency.

| Attention | KV Cache | Computation | | Expressivity |
| | | Attention | Linear | |
|---|---|---|---|---|
| **MHA** | 6 | 4 | 5 | Strong |
| **GQA** | 4 | 4 | 2 | Moderate |
| **MLA** | 2 | 5 | 6 | Strong |
| **GVA** | 5 | 3 | 4 | Moderate |
| **GHA** | 3 | 1 | 1 | Weak |
| **GTA (Ours)** | 1 | 2 | 3 | Strong |

GTA's KV cache scales as $(n_k d_h + n_c d_l)N$ compared to MHA's $2n_h d_h N$, where $n_k \ll n_h$ and $n_c \ll n_h$, yielding a reduction factor of approximately $\frac{2H}{n_k d_h + n_c d_l}$. The attention computation is similarly reduced from $2n_h d_h N^2$ to $n_q(d_h + d_l)N^2$, providing proportional inference speedups. While GTA introduces additional linear computation, this trade-off substantially improves model expressivity, rivaling MHA while maintaining efficiency comparable to other efficient variants.

## 5.2 CONDUCTING EMPIRICAL BENCHMARKS

To substantiate the theoretical advantages, we benchmark GTA-1B against GQA-1B and MLA-1B using the `LLM-Viewer` (Yuan et al., 2024) framework on an NVIDIA H100 80GB GPU. This framework simulates optimal inference performance based on hardware specifications and model configurations. Figure 4 illustrates the prefill and decode times across various configurations. GTA-1B consistently outperforms both GQA-1B and MLA-1B in compute-bound prefill and I/O-bound decode phases, demonstrating superior latency characteristics.

We further validate GTA's effectiveness across diverse settings: (1) **Multi-device evaluation** on NVIDIA H100-PCIe, A100, and A100-40G shows consistent efficiency gains; (2) **Long-context scaling** up to 128K tokens demonstrates that GTA's advantages become more pronounced with increasing sequence length; (3) **Model scaling** to 8B parameters confirms that performance improvements are maintained at larger model sizes. More hardware configurations and detailed evaluation results are provided in Appendix C.

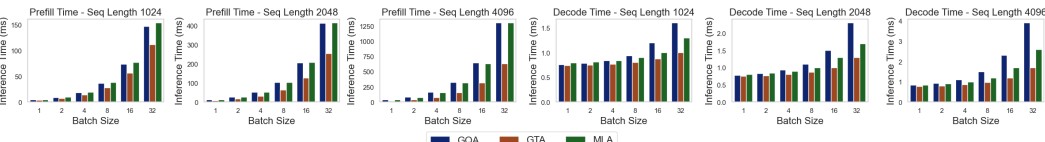

Figure 4: Prefill and decode times for GTA-1B, MLA-1B and GQA-1B across configurations on an NVIDIA H100 80GB GPU. GTA-1B achieves lower latency in both compute-bound prefill and I/O-bound decode phases, showcasing its enhanced efficiency.

## 5.3 REAL-WORLD DEPLOYMENT PERFORMANCE

Following PLM (Deng et al., 2025), we evaluate GTA-1B's real-world performance through inference experiments using the `torch` library. We measure prefill and decode times across diverse hardware platforms: NVIDIA H100 (server-grade GPU), NVIDIA A800 (server-grade GPU), RTX 3060 (consumer-grade GPU), Apple M2 (ARM-based processor), and BCM2712 (mobile processor). This approach captures hardware-specific optimizations and system-level overheads, providing direct measurements of real-world inference latency beyond theoretical simulations from `LLM-Viewer`.

We customize batch sizes to reflect realistic usage scenarios: server-grade GPUs (H100, A800) use prefill batch size 32 and decode batch size 64 for high-throughput environments; consumer devices

(M2, BCM2712) use batch size 1 for individual users; RTX 3060 uses prefill batch size 4 and decode batch size 16 for moderate workloads.

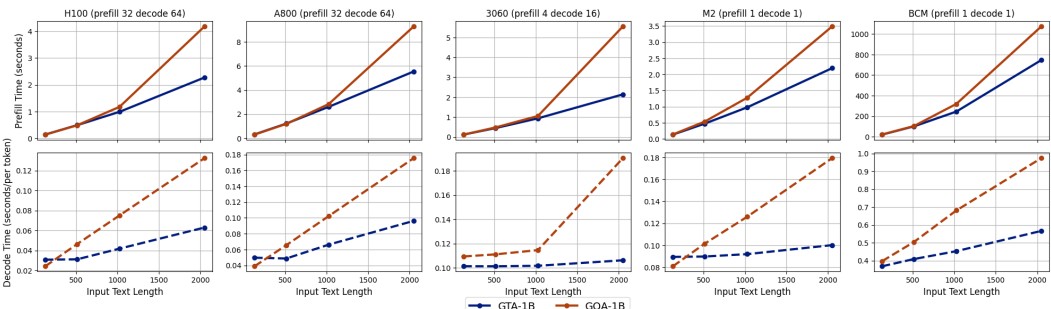

Figure 5: Comparison of prefill (top row) and decode (bottom row) times for GTA-1B and GQA-1B across various configurations on NVIDIA H100, NVIDIA A800, RTX 3060, Apple M2, and BCM2712. Prefill plots (top) display input text length on the x-axis and time required on the y-axis. Decode plots (bottom) show starting generation length on the x-axis and time to generate 128 tokens on the y-axis.

As shown in Figure 5, GTA-1B (blue solid line) consistently outperforms GQA-1B (orange dashed line) across all platforms. The performance advantage is particularly pronounced at longer input lengths (e.g., 2k tokens) and during extended generation phases, demonstrating GTA-1B's robustness across diverse hardware configurations.

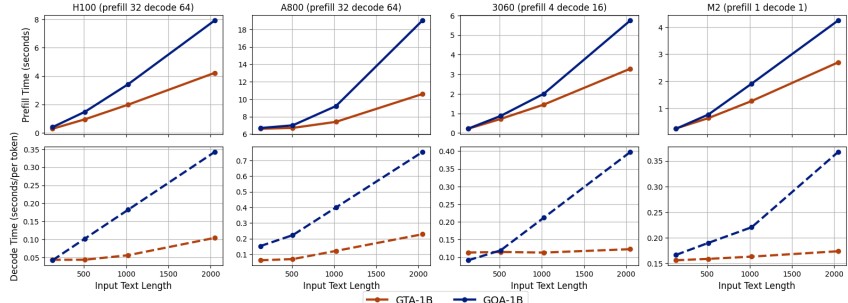

Figure 6: Performance comparison of GTA-1B and GQA-1B with cache offload enabled, showing prefill (top row) and decode (bottom row) times across different hardware configurations. Cache offload transfers the key-value cache to CPU memory to alleviate GPU memory constraints, resulting in I/O-bound conditions due to frequent data transfers.

Figure 6 demonstrates performance with cache offload enabled. GTA-1B maintains its advantages in I/O-bound scenarios where frequent data transfers occur between GPU and CPU memory, with particularly notable improvements in decode times across all platforms.

GTA-1B consistently surpasses GQA-1B in both prefill and decode performance across all hardware platforms, with significant advantages at longer input lengths. Its superior performance in both standard and I/O-bound conditions demonstrates practical applicability for server-grade and consumer-grade deployments, enhancing attention mechanism efficiency through reduced computational complexity and memory demands. Further experimental details, including comprehensive hardware specifications, are provided in Appendix C.4.

## 6    CONCLUSION

We present Grouped-head Latent Attention (GTA), which shares attention maps across heads and encodes values in a learned latent space to exploit redundancy. GTA reduces attention FLOPs by up to 62.5% and reduce KV cache size by up to 70% compared to GQA, matching perplexity while doubling inference speed on commodity hardware. By seeking the trade-off between efficiency and expressivity, GTA enables efficient LLM design and deployments across a wide range of real-world scenarios. The limitation stems from our lack of engineering-focused optimization efforts, which prevents us from achieving the theoretical upper bound of efficiency gains.

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

# A    TRAINING DETAIL

## A.1    PRETRAIN DETAIL

This section provides a comprehensive overview of the pretraining configurations and procedures employed in our experiments. We detail the model hyperparameters, data settings, and training specifics to ensure reproducibility and provide further insights into our methodology. The experiments were conducted on 4 nodes, each equipped with 8 NVIDIA A800 GPUs (80GB memory), totaling 32 GPUs for distributed training.

**Hardware Configuration**    Our training infrastructure consisted of 4 computing nodes, with each node containing 8 NVIDIA A800 GPUs (80GB memory). The distributed training setup allowed flexible allocation of GPU resources, scaling from single-node (8 GPUs) to full-cluster (32 GPUs) configurations depending on model size and training requirements.

**Model hyperparameters**    The key architectural hyperparameters for our models are summarized in Table 5 and Table 6 . We present configurations for 160M, 500M, 1B and 8B parameter models, highlighting the variations across different attention mechanisms: MHA, MLA, GQA, and our proposed GTA variants.

Table 5: Model hyperparameters for 160MB and 500MB

|  | 160M | | | | | 500M | | | | |
|---|---|---|---|---|---|---|---|---|---|---|
|  | **MHA** | **MLA** | **GQA** | **GTA1** | **GTA2** | **MHA** | **MLA** | **GQA** | **GTA3** | **GTA4** |
| **Number of layers** | 24 | 24 | 24 | 24 | 24 | 24 | 24 | 24 | 24 | 24 |
| **Hidden Dimension** | 768 | 768 | 768 | 768 | 768 | 1280 | 1280 | 1280 | 1280 | 1280 |
| **Intermediate Size** | 1920 | 1920 | 1920 | 1920 | 1920 | 3584 | 3584 | 3584 | 3584 | 3584 |
| **Number of Attention Heads** | 12 | 12 | 12 | 12 | 12 | 20 | 20 | 20 | 20 | 20 |
| **Number of Q Heads** | 12 | 12 | 12 | 3 | 6 | 20 | 20 | 20 | 5 | 10 |
| **Numbern of V Heads** | 12 | 1 | 3 | 1 | 1 | 20 | 1 | 4 | 1 | 2 |
| **Numbern of K Heads** | 12 | 1 | 3 | 1 | 1 | 20 | 1 | 4 | 1 | 1 |
| **KV Lora Rank** | — | 256 | — | — | — | — | 320 | — | — | — |
| **Compressed V Head Dimension** | — | — | — | 128 | 128 | — | — | — | 128 | 128 |
| **Vocabulary Size** | 32000 | 32000 | 32000 | 32000 | 32000 | 32000 | 32000 | 32000 | 32000 | 32000 |
| **Activation Function** | silu | silu | silu | silu | silu | silu | silu | silu | silu | silu |
| **Tie Embeddinng** | TRUE | TRUE | TRUE | TRUE | TRUE | FALSE | FALSE | FALSE | FALSE | FALSE |
| **Params(M)** | 178.78 | 172.54 | 158.50 | 160.75 | 164.13 | 543.27 | 516.00 | 483.23 | 486.98 | 500.11 |

Table 6: Model hyperparameters for 1B and 8B

|  | 1B | | | 8B | | |
|---|---|---|---|---|---|---|
|  | **MLA-1B** | **GQA-1B** | **GTA-1B** | **MLA-8B** | **GQA-8B** | **GTA-8B** |
| **Number of layers** | 54 | 54 | 54 | 32 | 32 | 32 |
| **Hidden Dimension** | 1280 | 1280 | 1280 | 4096 | 4096 | 4096 |
| **Intermediate Size** | 3584 | 3584 | 3584 | 14336 | 14336 | 14336 |
| **Number of Attention Heads** | 20 | 20 | 20 | 32 | 32 | 32 |
| **Number of Q Heads** | 20 | 20 | 5 | 32 | 32 | 8 |
| **Numbern of V Heads** | 1 | 5 | 1 | 1 | 8 | 2 |
| **Numbern of K Heads** | 1 | 5 | 1 | 1 | 8 | 1 |
| **KV Lora Rank** | 320 | - | - | 512 | - | - |
| **Compressed V Head Dimension** | - | - | 128 | - | - | 256 |
| **Vocabulary Size** | 128256 | 128256 | 128256 | 128256 | 128256 | 128256 |
| **Activation Function** | silu | silu | silu | silu | silu | silu |
| **Tie Embeddinng** | TRUE | TRUE | TRUE | FALSE | FALSE | FALSE |

**Data and hyperparameters**    Table 7 details the key hyperparameters used in our pretraining experiments. We employed two different scaling configurations, referred to as "Validation" and "Scaling", to assess the impact of model and data scaling on performance. These configurations differ primarily in global batch size, learning rate, and certain Adam optimizer settings.

Table 7: Experiments hyperparameters.

| Hyperparameter | Validation | Scaling | SFT |
|---|---|---|---|
| Global Batch Size | 800 | 2048 | 96 |
| Learning Rate | 2.00E-04 | 1.00E-03 | 2.00E-5 |
| Learning Rate Scheduler | cosine | consine | cosine |
| Warm up rate | 0.01 | 0.01 | 0.1 |
| Weight Decay | default(0.0) | 0.1 | 0.1 |
| Adam $\beta_1$ | default(0.9) | 0.9 | 0.9 |
| Adam $\beta_2$ | default(0.999) | 0.95 | 0.95 |
| Clip Grad | 1.0 | 1.0 | 1.0 |
| Rms Norm Eps | default(1e-06) | 1e-5 | 1e-5 |
| Attention Dropout | 0 | 0 | 0 |
| Hidden Dropout | 0 | 0 | 0 |
| Epoch | 1 | 1 | 4 |

## A.2 LOSS CURVE

To provide insights into the training dynamics, we present the loss curves for various model configurations. Figure 7, Figure 8, Figure 9 and Figure 10 illustrate the training and evaluation loss trajectories for the 160M and 500M models across different sequence lengths. Notably, the evaluation loss is slightly lower than the training loss, which can be attributed to the evaluation being conducted on a subset of the data for efficiency, potentially comprising a simpler distribution.

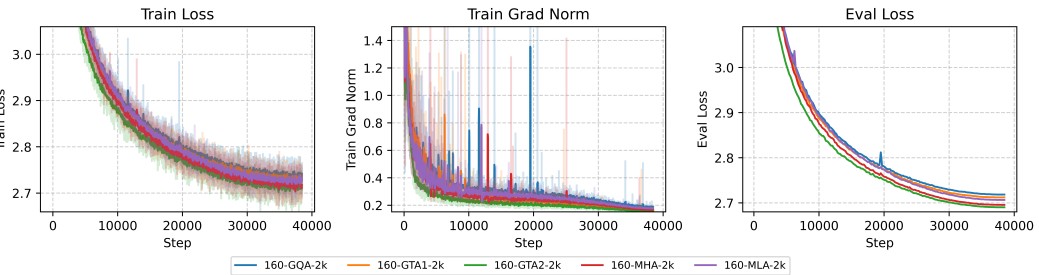

Figure 7: Loss Curve for 160M with 2048 sequence length

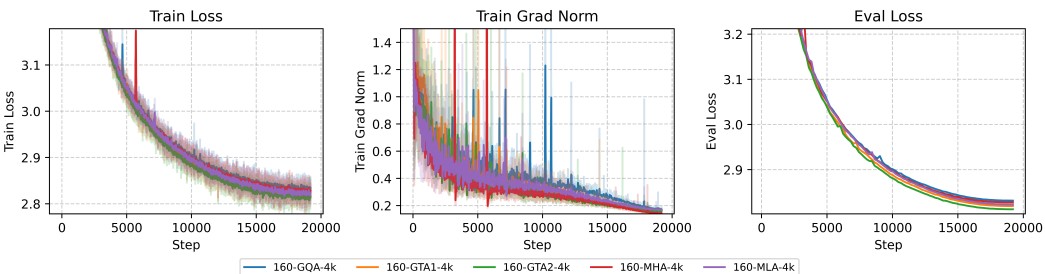

Figure 8: Loss Curve for 160M with 4096 sequence length

## A.3 SFT DETAIL

In the SFT stage, we trained our model using the `tulu-3-sft-mixture` Lambert et al. (2024) dataset. We utilized the LlamaFactory Zheng et al. (2024) framework with nearly all default hyperparameters. Additional training details are available in Table 7.

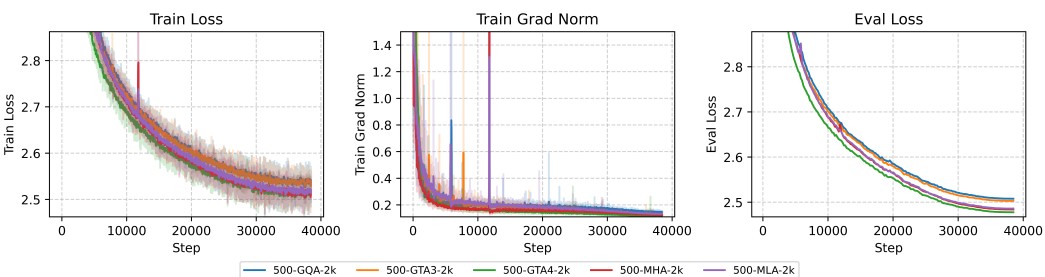

Figure 9: Loss Curve for 500M with 2048 sequence length

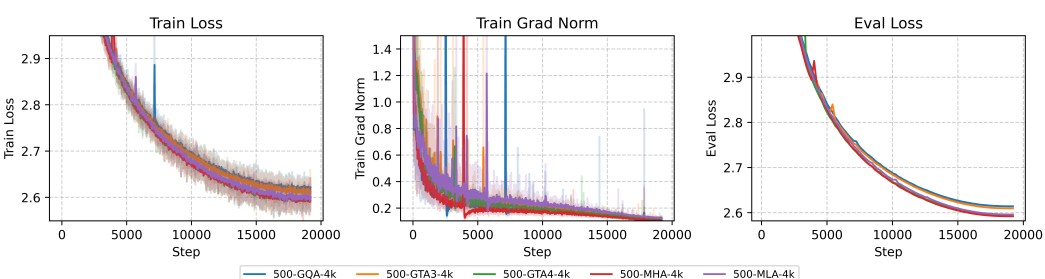

Figure 10: Loss Curve for 500M with 4096 sequence length

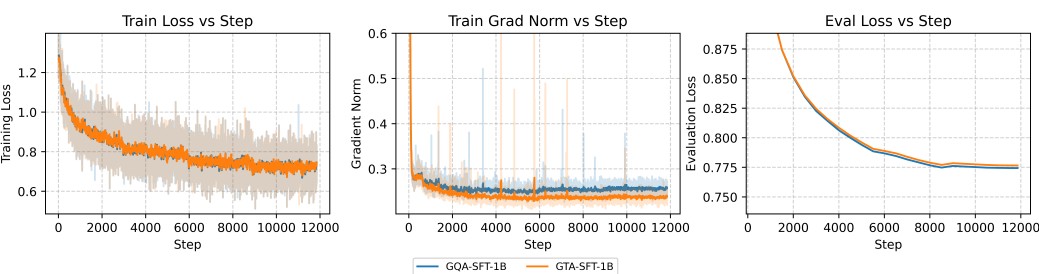

Figure 11: Loss curve for SFT

## A.4 SENSITIVITY ANALYSIS RESULT

**Ablation study on GTA components**  We provide comprehensive results and configurations for the ablation studies on the GTA components. All experiments were conducted on models with 500M parameters and a sequence length of 2048. The base configuration uses a standard transformer architecture, with variations introduced by enabling or disabling the Shared Attention (SA), Nonlinear decoding (NL), and Up-projection (UP) components.

Table 8 presents detailed results, including evaluation loss, perplexity on Wikitext, and accuracy on downstream tasks such as PIQA, HellaSwag, ARC-easy (ARC-e), ARC-challenge (ARC-c), and Winogrande. The average (Avg) is computed across the downstream tasks.

Table 8: Ablation study on GTA components.

| Model | Eval Loss | Wikitext PPL | PIQA | HellaSwag | ARC-e | ARC-c | Winogrande | Avg |
|---|---|---|---|---|---|---|---|---|
| MHA | 2.486 | 17.58 | 68.64 | **35.11** | 47.85 | 20.83 | 50.98 | 44.68 |
| GTA | **2.475** | 16.87 | 68.51 | 34.98 | 47.45 | 21.07 | **53.31** | 45.06 |
| GTA w/o UP | 2.483 | 17.13 | 68.44 | 34.28 | 46.88 | 21.85 | 52.22 | 44.73 |
| GTA w/o SA & UP | 2.479 | **16.51** | **68.68** | 34.96 | **47.95** | **22.46** | 52.96 | **45.40** |
| GTA w/o NL & UP | 2.521 | 19.05 | 67.56 | 34.12 | 45.44 | 19.05 | 51.42 | 43.52 |
| GQA (GTA w/o SA & NL & UP) | 2.508 | 18.52 | 68.61 | 34.31 | 46.72 | 22.44 | 51.62 | 44.73 |

From the results, the full GTA achieves the lowest evaluation loss and a balanced improvement across metrics. Removing NL leads to the most significant degradation, highlighting its importance for performance. SA provides efficiency benefits (as seen in latency reductions in the main text), while UP offers minor gains without substantial overhead. These findings validate the synergistic effects of the GTA components.

**Impact of Shared Attention Matrix**  To understand the importance of sharing attention matrix across heads in our GTA architecture, we conduct an ablation study comparing shared vs. non-shared attention matrix. As shown in Table 9, while sharing attention matrix reduces the parameter count from 511.37M to 492.61M, it actually improves performance slightly (2.4995 vs. 2.496). This suggests that our approach not only saves memory and computation but also provides a beneficial regularization effect, supporting the hypothesis that traditional attention mechanisms may be over-parameterized.

Table 9: Ablation study on the effect of sharing attention matrix in GTA models (500M parameter range).

| Configuration | Parameters | Eval Loss | Cache/layer | Seq Length |
|---|---|---|---|---|
| GTA with 5 attention matrix groups | 486.98M | 2.5031 | 192 (64+128) | 2048 |
| GTA with 10 attention matrix groups | 492.61M | 2.4995 | 192 (64+128) | 2048 |
| GTA without attention matrix groups | 511.37M | **2.4960** | 192 (64+128) | 2048 |

**Effect of Head Dimension**  We also investigate the effect of increasing the head dimension while keeping the total parameter count similar. Table 11 compares models with head dimensions of 64 and 128. Doubling the head dimension improves performance in both GQA and GTA models, with GTA consistently outperforming GQA. Notably, GTA with doubled head dimensions achieves our best performance (2.492), suggesting that allocating more capacity to each head while sharing attention matrixs is an effective design choice for attention mechanisms.

Table 10: Ablation study on the effect of head dimension in GQA and GTA models (500M parameter range).

| Model | Head Dim | Parameters | Head Dim | Eval Loss | Cache/layer | Seq Length |
|---|---|---|---|---|---|---|
| GQA | 64 | 483.23M | 64 | 2.5079 | 512 (4×2×64) | 2048 |
| GTA | 64 | 492.61M | 64 | **2.4995** | 192 (64+128) | 2048 |
| GQA | 128 | 483.23M | 128 | 2.5038 | 512 (2×2×128) | 2048 |
| GTA | 128 | 498.24M | 128 | **2.4844** | 384 (128+256) | 2048 |

**Choice of Nonlinearity**   We explored different nonlinear activation functions for the gating mechanism, including ReLU$^2$, Silu, and Sigmoid, and observed that performance degrades as the sparsity of the activation increases. Sigmoid, with its smooth and bounded output, consistently outperformed sparser alternatives like Silu and ReLU$^2$, which introduce more zeros and reduce the effective rank of the value representation. This behavior contrasts with typical MLP architectures, where sparse activations like ReLU often enhance performance by promoting feature selectivity. In GTA, however, the reduced rank caused by sparsity impairs the expressivity of value, underscoring the importance of maintaining a higher rank in the value representation for effective attention computation.

Table 11: Ablation study on the effect of activation function in GTA models (500M parameter range).

| Model | Parameters | Activation | Eval Loss | Cache/layer | Seq Length |
|-------|-----------|-----------|-----------|-------------|------------|
| GTA | 492.61M | Sigmoid | **2.4995** | 192 (64+128) | 2048 |
| GTA | 492.61M | Silu | 2.5314 | 192 (64+128) | 2048 |
| GTA | 492.61M | ReLU$^2$ | 2.5502 | 192 (64+128) | 2048 |

## B COMPUTATIONAL ANALYSIS

### B.1 THEORETICAL EFFICIENCY ANALYSIS

Table 12 compares the key-value (KV) cache size and computational complexity across attention mechanisms. GTA achieves a KV cache size of $(n_k d_h + n_c d_l)N$, significantly smaller than MHA's $2n_h d_h N$. Its attention computation, $n_q(d_h + d_l)N^2$, is also lower than MHA's $2n_h d_h N^2$, enhancing inference efficiency. While GTA introduces additional linear computation, this trade-off substantially improves model expressivity, rivaling MHA while maintaining efficiency comparable to GVA and GHA.

Table 12: Comparison of computational complexity and memory requirements for different attention mechanisms. $H$ is the hidden dimension, $N$ is the sequence length, $n_q, n_k, n_v, n_c$ are the number of query, key, value, and latent value heads, respectively, $d_h$ is the per-head dimension, and $d_l$ is the latent dimension.

| Attention | KV Cache per Layer | Computation per Layer | | Expressivity |
|---|---|---|---|---|
| | | Attention | Linear | |
| **MHA** | $2n_h d_h N$ | $2n_h d_h N^2$ | $4NH^2$ | Strong |
| GQA | $2n_k d_h N$ | $2n_h d_h N^2$ | $2NH^2 + 2n_k d_h NH$ | Moderate |
| MLA | $(d_c + d_{rope})N$ | $n_h(d_{rope} + 2d_{nope})N^2$ | $\left((d_c + d_{rope})H + n_h(d_{rope} + d_{nope})H + 2n_h d_l d_{nope} + H^2\right)N$ | Strong |
| GVA | $(H + n_k d_h)N$ | $(n_q d_h + n_h d_h)N^2$ | $2NH^2 + 2n_k d_h NH$ | Moderate |
| GHA | $(n_k d_h + n_v d_h)N$ | $(n_q d_h + n_h d_h)N^2$ | $NH^2 + n_q d_h NH + n_k d_h NH + n_v d_h NH$ | Weak |
| **GTA (Ours)** | $\mathbf{(n_k d_h + n_c d_l)N}$ | $\mathbf{n_q(d_k + d_l)N^2}$ | $\mathbf{2NH^2 + (n_q d_h + n_k d_h + n_c d_l + d_l)NH}$ | Strong |

As shown in Table 12, GTA achieves substantial efficiency gains in both computation and memory usage. The KV cache size is reduced from $2HN$ in MHA to $(n_k d_h + n_c d_l)N$, where $n_k \ll n_h$ and $n_c \ll n_h$. This translates to a reduction factor of approximately $\frac{2H}{n_k d_h + n_c d_l}$, which can be significant for large models. The attention computation is also reduced from $2n_h d_h N^2$ to $n_q(d_h + d_l)N^2$, offering proportional speedups during inference.

### B.2 GTA

#### B.2.1 DEFINITION

Let $\boldsymbol{h}_t \in \mathbb{R}^H$ represent the input hidden state for the $t$-th token in the attention mechanism. The grouped key and compressed value for the $j$-th head are denoted by $\boldsymbol{k}_{t,j} \in \mathbb{R}^{d_h}$ and $\boldsymbol{c}_{t,j} \in \mathbb{R}^{d_c}$, respectively. The position-independent query for the $k$-th head is represented as $\boldsymbol{q}_{t,k} \in \mathbb{R}^{d_h}$. The computations for the attention mechanism proceed as follows:

$$\boldsymbol{k}_{t,j} = \text{RoPE}\left(W_{\text{K},j}\boldsymbol{h}_t\right),$$
$$\boldsymbol{q}_{t,k} = \text{RoPE}\left(W_{\text{Q},k}\boldsymbol{h}_t\right),$$
$$\boldsymbol{v}_{t,j}^C = W_{\text{V},j}\boldsymbol{h}_t,$$

where $W_{\text{K},j} \in \mathbb{R}^{d_h \times d}$ and $W_{\text{C},j} \in \mathbb{R}^{d_h \times d}$ are the up-projection matrices for grouped key and compressed value for the $j$-th kv head, and $W_{\text{Q},k} \in \mathbb{R}^{d_h \times d}$ for the $k$-th head, respectively.

The attention outputs $\{\boldsymbol{o}_{t,i}\}$ are calculated as follows:

$$\boldsymbol{o}_{t,i} = \left(\sum_{k=1}^{t} \text{Softmax}_k\left(\frac{\boldsymbol{q}_{t,Q(i)}^\top \boldsymbol{k}_{k,K(i)}}{\sqrt{d_h}}\right)\boldsymbol{v}_{k,V(i)}\right)W_{P,i},$$

where $W_{P,i} \in \mathbb{R}^{d_h \times d_c}$ is the attention projection matrix.

The $i$-th head gate $\boldsymbol{g}_{t,i} \in \mathbb{R}^{d_h \times H}$ is proceed as follows:

$$\boldsymbol{g}_{t,i} = W_{\text{G},i}\boldsymbol{h}_t,$$

The final output is obtained by combining the attention results from all heads through a linear projection:

$$\boldsymbol{u}_t = W_{\mathrm{O}}\left[\boldsymbol{o}_{t,1} \odot \boldsymbol{g}_{t,1}; \boldsymbol{o}_{t,2} \odot \boldsymbol{g}_{t,2}; \ldots ; \boldsymbol{o}_{t,n_h} \odot \boldsymbol{g}_{t,n_h}\right],$$

where $W_{\mathrm{O}} \in \mathbb{R}^{d \times d_{\mathrm{nope}} n_h}$ is the output projection matrix and $n_h$ is the number of attention heads.

### B.2.2 PREFILL

For an input sequence of length $N$, the computational complexity begins with the projection operations for keys $\boldsymbol{k}_{t,j}$ and compressed values $\boldsymbol{c}_{t,j}$, requiring $\mathcal{O}(d_{\mathrm{h}} N H)$ and $\mathcal{O}(d_{\mathrm{c}} N H)$ operations. The query projection $\boldsymbol{q}_{t,k}$ further contributes $\mathcal{O}(d_{\mathrm{h}} N H)$. The gate projection requires $\mathcal{O}(N H^2)$ and the attention projection for each head requires $\mathcal{O}(d_h d_c N)$. The output projection requires $\mathcal{O}(N H^2)$.

Respectively. Aggregating these components, the total linear projection cost becomes:

$$\mathcal{O}\Big(2 N H^2 + (n_{\mathrm{q}} d_{\mathrm{h}} + n_{\mathrm{k}} d_{\mathrm{h}} + n_{\mathrm{v}} d_{\mathrm{c}} + d_c) N H\Big).$$

The attention mechanism's computational complexity arises from pairwise interactions between sequence elements, resulting in a quadratic scaling with sequence length $N$. Computing attention scores $QK^T$ has a complexity of $\mathcal{O}(n_q d_h N^2)$. Generating the attention output by values $V$ adds $\mathcal{O}(n_q d_c N^2)$. The total complexity is thus $\mathcal{O}(n_q (d_h + d_c) N^2)$.

Combining all terms, the total computational complexity for the prefill phase is:

$$\mathrm{Prefill}_{GTA} = \mathcal{O}\Big(2 N H^2 + (n_{\mathrm{q}} d_{\mathrm{h}} + n_{\mathrm{k}} d_{\mathrm{h}} + n_{\mathrm{v}} d_{\mathrm{c}} + d_c) N H + n_q (d_h + d_c) N^2\Big).$$

### B.2.3 DECODE

For an input sequence of length $N - 1$, the decoder phase computes the $N$-th token's representations through successive transformations. Key and value projections $\boldsymbol{k}_{N,j}$ and $\boldsymbol{c}_{N,j}$ require $\mathcal{O}(d_{\mathrm{h}} H)$ and $\mathcal{O}(d_{\mathrm{c}} H)$ operations, while the query projection $\boldsymbol{q}_{N,i}$ incurs $\mathcal{O}(d_{\mathrm{h}} H)$. The gate projection requires $\mathcal{O}(H^2)$ and the attention projection for each head requires $\mathcal{O}(d_h d_c)$. The output projection requires $\mathcal{O}(H^2)$. The total computational linear projection cost:

$$\mathcal{O}\Big(2 H^2 + (n_{\mathrm{q}} d_{\mathrm{h}} + n_{\mathrm{k}} d_{\mathrm{h}} + n_{\mathrm{v}} d_{\mathrm{c}} + d_c) H\Big).$$

The attention mechanism, operating over cached historical states, scales as $\mathcal{O}(2 n_h d_{\mathrm{h}} N)$, reflecting linear dependence on sequence length $N$. Aggregating all components, the total computational cost is:

$$\mathrm{Generate}_{GTA} = \mathcal{O}\Big(2 H^2 + (n_{\mathrm{q}} d_{\mathrm{h}} + n_{\mathrm{k}} d_{\mathrm{h}} + n_{\mathrm{v}} d_{\mathrm{c}} + d_c) H + 2 n_h d_{\mathrm{h}} N\Big).$$

Caching historical keys $\{\boldsymbol{k}_{t,j}\}$ and values $\{\boldsymbol{v}_{t,j}\}$ for $t = 1, \ldots, N - 1$ demands memory:

$$\mathrm{Cache}_{GTA} = (n_k d_h + n_v d_c) N,$$

## B.3 MLA

### B.3.1 DEFINITION

Let $\boldsymbol{h}_t \in \mathbb{R}^H$ represent the input hidden state for the $t$-th token in the attention mechanism. The low-rank key-value joint compression state is denoted as $\boldsymbol{c}_t^{KV} \in \mathbb{R}^{d_c}$, while the decompressed key and value for the $i$-th head are denoted by $\boldsymbol{k}_{t,i}^{C} \in \mathbb{R}^{d_{\mathrm{nope}}}$ and $\boldsymbol{v}_{t,i}^{C} \in \mathbb{R}^{d_{\mathrm{nope}}}$, respectively. The position-independent query for the $i$-th head is represented as $\boldsymbol{q}_{t,i}^{C} \in \mathbb{R}^{d_{\mathrm{nope}}}$. The computations for the attention mechanism proceed as follows:

$$\boldsymbol{c}_t^{KV} = W_{\text{DKV}}\boldsymbol{h}_t,$$

$$\boldsymbol{k}_{t,i}^{C} = W_{\text{UK},i}\boldsymbol{c}_t^{KV},$$

$$\boldsymbol{k}_t^{R} = \text{RoPE}\left(W_{\text{KR}}\boldsymbol{h}_t\right),$$

$$\boldsymbol{k}_{t,i} = \left[\boldsymbol{k}_{t,i}^{C}; \boldsymbol{k}_t^{R}\right],$$

$$\boldsymbol{q}_{t,i}^{C} = W_{\text{Q},i}\boldsymbol{h}_t,$$

$$\boldsymbol{q}_{t,i}^{R} = \text{RoPE}\left(W_{\text{QR},i}\boldsymbol{h}_t\right),$$

$$\boldsymbol{q}_{t,i} = \left[\boldsymbol{q}_{t,i}^{C}; \boldsymbol{q}_{t,i}^{R}\right],$$

$$\boldsymbol{v}_{t,i}^{C} = W_{\text{UV},i}\boldsymbol{c}_t^{KV},$$

where $W_{\text{DKV}} \in \mathbb{R}^{d_c \times H}$ is the down-projection matrix for key-value compression, $W_{\text{UK},i} \in \mathbb{R}^{d_{\text{nope}} \times d_c}$ and $W_{\text{UV},i} \in \mathbb{R}^{d_{\text{nope}} \times d_c}$ are the up-projection matrices for decompressed key and value for the $i$-th head, $W_{\text{KR}} \in \mathbb{R}^{d_{\text{rope}} \times H}$ generates the shared positional key component via RoPE Su et al. (2024), and $W_{\text{Q},i} \in \mathbb{R}^{d_{\text{nope}} \times H}$ and $W_{\text{QR}} \in \mathbb{R}^{d_{\text{rope}} \times H}$ generate the position-independent and RoPE-enhanced query components for the $i$-th head.

The attention outputs $\{\boldsymbol{o}_{t,i}\}$ are calculated as follows:

$$\boldsymbol{o}_{t,i} = \sum_{j=1}^{t} \text{Softmax}_j\left(\frac{\boldsymbol{q}_{t,i}^{\top}\boldsymbol{k}_{j,i}}{\sqrt{d_h}}\right)\boldsymbol{v}_{j,i}^{C},$$

where $d_h = d_{\text{nope}} + d_{\text{rope}}$ represents the total head dimension. The final output is obtained by combining the attention results from all heads through a linear projection:

$$\boldsymbol{u}_t = W_{\text{O}}\left[\boldsymbol{o}_{t,1}; \boldsymbol{o}_{t,2}; \ldots; \boldsymbol{o}_{t,n_h}\right],$$

where $W_{\text{O}} \in \mathbb{R}^{H \times d_{\text{nope}}n_h}$ is the output projection matrix and $n_h$ is the number of attention heads.

### B.3.2 PREFILL

Let the input sequence length be $N$. The computational complexity for projecting the context vector $\boldsymbol{c}_t^{KV}$ is $\mathcal{O}(d_c N H)$. Subsequent projections for content-based keys $\boldsymbol{k}_{t,i}^{C}$ and values $\boldsymbol{v}_{t,i}^{C}$ require $\mathcal{O}(2d_c d_{\text{nope}} N)$ operations, while the query projection $\boldsymbol{q}_{t,i}^{C}$ incurs $\mathcal{O}(d_{\text{nope}} N H)$. For rotary position embeddings (RoPE), the projections for $\boldsymbol{k}_t^{R}$ and $\boldsymbol{q}_{t,i}^{R}$ each demand $\mathcal{O}(d_{\text{rope}} N H)$. The output projection further adds $\mathcal{O}(N H^2)$.

The total computational linear projection cost for generating keys $\{\boldsymbol{k}_{t,i}\}$, queries $\{\boldsymbol{q}_{t,i}\}$, values $\{\boldsymbol{v}_{t,i}\}$ and outputs $\boldsymbol{o}_t$ combines these components:

$$\mathcal{O}\left((d_c + d_{\text{rope}})NH + n_h(d_{\text{nope}} + d_{\text{rope}})NH + 2n_h d_c d_{\text{nope}}N + NH^2\right).$$

The attention mechanism's computational complexity arises from pairwise interactions between sequence elements, resulting in a quadratic scaling with sequence length $N$. Computing attention scores $QK^T$ has a complexity of $\mathcal{O}(n_h(d_{\text{rope}} + d_{\text{nope}})N^2)$. Generating the attention output by values $V$ adds $\mathcal{O}(n_h d_{\text{nope}} N^2)$. The total complexity is thus $\mathcal{O}(n_h(d_{\text{rope}} + 2d_{\text{nope}})N^2)$.

Aggregating all terms, the overall computational complexity becomes:

$\text{Prefill}_{mla} =$

$\mathcal{O}\left((d_c + d_{\text{rope}})NH + n_h(d_{\text{nope}} + d_{\text{rope}})NH + 2n_h d_c d_{\text{nope}}N + NH^2 + n_h(d_{\text{rope}} + 2d_{\text{nope}})N^2\right)$

### B.3.3    DECODE

Consider an input sequence of length $N - 1$. The computational complexity to generate the $N$-th token's joint compression state $\boldsymbol{c}_N^{KV}$ is $\mathcal{O}(d_c H)$. Subsequent projections for the rotary position embedding (RoPE)-based key $\boldsymbol{k}_N^R$ and query $\boldsymbol{q}_{N,i}^R$ each require $\mathcal{O}(d_{\text{rope}} H)$, while the content-based query $\boldsymbol{q}_{N,i}^C$ incurs $\mathcal{O}(d_{\text{nope}} H)$. For historical tokens $t = 1, \ldots, N$, the projections of content-based keys $\{\boldsymbol{k}_{t,i}^C\}$ and values $\{\boldsymbol{v}_{t,i}^C\}$ scale as $\mathcal{O}(2 d_c d_{\text{nope}} N)$, while the output projection requires $\mathcal{O}(H^2)$. The total computational linear projection cost:

$$\mathcal{O}\left((d_c + d_{\text{rope}})H + n_h(d_{\text{nope}} + d_{\text{rope}})H + 2 n_h d_c d_{\text{nope}} N + H^2\right).$$

The attention mechanism's computational complexity arises from pairwise interactions between sequence elements. Computing attention scores $QK^T$ has a complexity of $\mathcal{O}(n_h(d_{\text{rope}} + d_{\text{nope}})N)$. Generating the attention output by values $V$ adds $\mathcal{O}(n_h d_{\text{nope}} N)$. The total complexity is thus $\mathcal{O}(n_h(d_{\text{rope}} + 2 d_{\text{nope}})N)$. Combining these components, the total computational cost is:

$$\text{Generate}_{\text{mla}} =$$
$$\mathcal{O}\left((d_c + d_{\text{rope}})H + n_h(d_{\text{nope}} + d_{\text{rope}})H + 2 n_h d_c d_{\text{nope}} N + H^2 + (n_h(d_{\text{rope}} + 2 d_{\text{nope}})N)\right).$$

Caching mechanisms store the joint compression states $\{\boldsymbol{c}_t^{KV}\}_{t=1,\ldots,N-1}$ and RoPE keys $\{\boldsymbol{k}_t^R\}_{t=1,\ldots,N-1}$, with memory footprint:

$$\text{Cache}_{\text{mla}} = (d_{\text{rope}} + d_c)N.$$

### B.4    GQA

### B.4.1    DEFINITION

Let $\boldsymbol{h}_t \in \mathbb{R}^H$ represent the input hidden state for the $t$-th token in the attention mechanism. The grouped key and value for the $j$-th kv head are denoted by $\boldsymbol{k}_{t,j} \in \mathbb{R}^{d_{\text{h}}}$ and $\boldsymbol{v}_{t,j} \in \mathbb{R}^{d_{\text{h}}}$, respectively. The position-independent query for the $i$-th head is represented as $\boldsymbol{q}_{t,i} \in \mathbb{R}^{d_{\text{h}}}$. The computations for the attention mechanism proceed as follows:

$$\boldsymbol{k}_{t,j} = \text{RoPE}\left(W_{\text{K},j} \boldsymbol{h}_t\right),$$
$$\boldsymbol{q}_{t,i} = \text{RoPE}\left(W_{\text{Q},i} \boldsymbol{h}_t\right),$$
$$\boldsymbol{v}_{t,j}^C = W_{\text{V},j} \boldsymbol{h}_t,$$

where $W_{\text{K},j} \in \mathbb{R}^{d_{\text{h}} \times H}$ and $W_{\text{V},j} \in \mathbb{R}^{d_{\text{h}} \times H}$ are the up-projection matrices for grouped key and value for the $j$-th kv head, and $W_{\text{Q},i} \in \mathbb{R}^{d_{\text{h}} \times H}$ for the $i$-th head, respectively.

The attention outputs $\{\boldsymbol{o}_{t,i}\}$ are calculated as follows:

$$\boldsymbol{o}_{t,i} = \sum_{k=1}^{t} \text{Softmax}_k \left(\frac{\boldsymbol{q}_{t,i}^\top \boldsymbol{k}_{k,i \bmod n_k}}{\sqrt{d_{\text{h}}}}\right) \boldsymbol{v}_{k,i \bmod n_k},$$

The final output is obtained by combining the attention results from all heads through a linear projection:

$$\boldsymbol{u}_t = W_{\text{O}}\left[\boldsymbol{o}_{t,1}; \boldsymbol{o}_{t,2}; \ldots; \boldsymbol{o}_{t,n_h}\right],$$

where $W_{\text{O}} \in \mathbb{R}^{H \times d_{\text{nope}} n_h}$ is the output projection matrix and $n_h$ is the number of attention heads.

### B.4.2   PREFILL

For an input sequence of length $N$, the computational complexity begins with the projection operations for keys $k_{t,j}$ and values $v_{t,j}$, each requiring $\mathcal{O}(2d_{\mathrm{h}}NH)$ operations. The query projection $q_{t,i}$ further contributes $\mathcal{O}(d_{\mathrm{h}}NH)$. The output projection requires $\mathcal{O}(H^2)$.

Respectively. Aggregating these components, the total linear projection cost becomes:

$$\mathcal{O}\Big(2NH^2 + 2n_{\mathrm{k}}d_{\mathrm{h}}NH\Big).$$

The attention mechanism's computational complexity arises from pairwise interactions between sequence elements, resulting in a quadratic scaling with sequence length $N$. Computing attention scores $QK^T$ has a complexity of $\mathcal{O}(n_h d_h N^2)$. Generating the attention output by values $V$ adds $\mathcal{O}(n_h d_h N^2)$. The total complexity is thus $\mathcal{O}(2n_h d_h N^2)$.

Combining all terms, the total computational complexity for the prefill phase is:

$$\mathrm{Prefill}_{gqa} = \mathcal{O}\Big(2NH^2 + 2n_{\mathrm{k}}d_{\mathrm{h}}NH + 2n_h d_h N^2\Big).$$

### B.4.3   DECODE

For an input sequence of length $N-1$, the decoder phase computes the $N$-th token's representations through successive transformations. Key and value projections $k_{N,j}$ and $v_{N,j}$ require $\mathcal{O}(2d_{\mathrm{h}}H)$ operations, while the query projection $q_{N,i}$ incurs $\mathcal{O}(d_{\mathrm{h}}H)$. The total computational linear projection cost:

$$\mathcal{O}\Big(2H^2 + 2n_{\mathrm{k}}d_{\mathrm{h}}H\Big).$$

The attention mechanism, operating over cached historical states, scales as $\mathcal{O}(2n_h d_{\mathrm{h}}N)$, reflecting linear dependence on sequence length $N$. Aggregating all components, the total computational cost is:

$$\mathrm{Generate}_{gqa} = \mathcal{O}\Big(2H^2 + 2n_{\mathrm{k}}d_{\mathrm{h}}H + 2n_h d_{\mathrm{h}}N\Big).$$

Caching historical keys $\{k_{t,j}\}$ and values $\{v_{t,j}\}$ for $t = 1, \ldots, N-1$ demands memory:

$$\mathrm{Cache}_{gqa} = 2n_{\mathrm{k}}d_{\mathrm{h}}N,$$

## C  EFFICIENCY ANALYSIS

### C.1  LONG-CONTEXT PERFORMANCE EVALUATION

To assess GTA's scalability for long-context applications, we evaluate performance across extended sequence lengths ranging from 4K to 128K tokens. As illustrated in Figure 12, our experiments on NVIDIA H100 80GB demonstrate that GTA's efficiency advantages become increasingly pronounced with longer sequences. The performance gap between GTA-1B and GQA-1B widens substantially as sequence length increases, with GTA showing superior latency characteristics across all tested configurations.

This scaling behavior aligns with our theoretical analysis, as the computational and memory efficiency improvements of GTA become more significant for longer sequences. The results confirm that GTA maintains its architectural benefits even under the demanding memory and computational requirements of extended context processing, making it particularly well-suited for applications requiring long-context understanding.

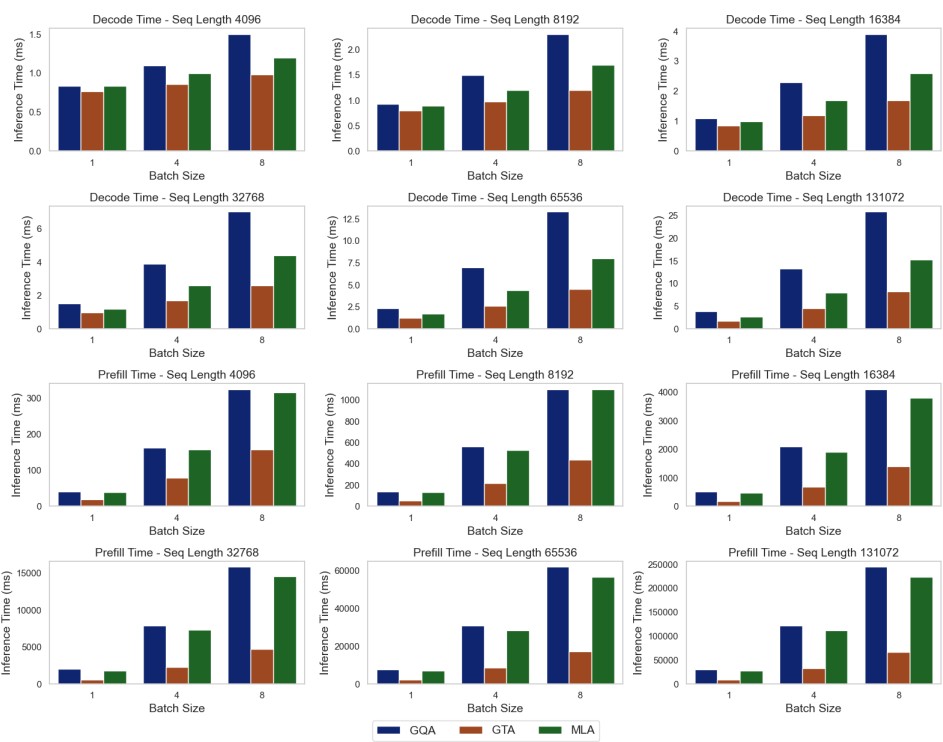

Figure 12: Performance comparison between GTA-1B, GQA-1B and MLA-1B across extended sequence lengths (4K-128K tokens) on NVIDIA H100 80GB GPU. GTA demonstrates increasing efficiency advantages as sequence length grows.

### C.2  MODEL SIZE SCALING EVALUATION

To validate that our approach scales beyond the 1B parameter regime, we conduct additional experiments with 8B parameter models. As shown in Figure 13, the GTA-8B model maintains the efficiency gains observed at smaller scales, demonstrating consistent performance improvements compared to GQA-8B across various configurations. This demonstrates that our architectural innovations remain effective as model capacity increases, suggesting promising applicability to larger language models.

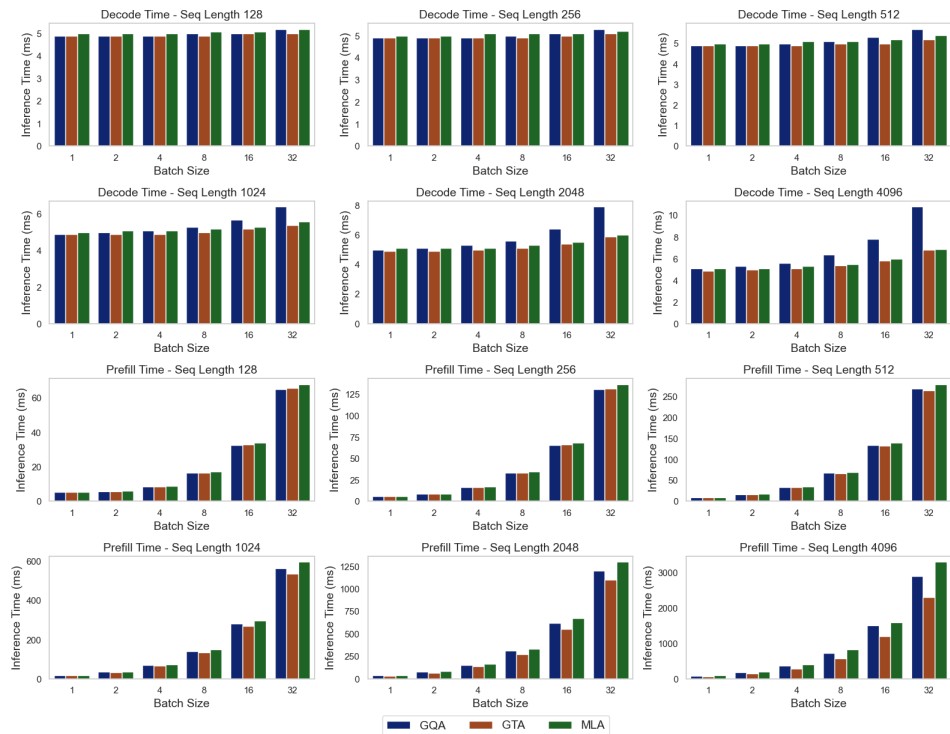

Figure 13: Performance comparison between GTA-8B, GQA-8B and MLA-8B across different configurations, demonstrating maintained efficiency advantages at larger model scales.

## C.3 MULTI-DEVICE EVALUATION

To comprehensively evaluate the robustness of GTA-1B's performance across diverse hardware platforms, we conducted extensive benchmarks using the `LLM-Viewer` framework, consistent with our main evaluation methodology. These experiments were performed on various NVIDIA GPUs, including NVIDIA A100 40GB, NVIDIA A100 80GB, NVIDIA H100 80GB, NVIDIA H100 PCIe 80GB. As presented in Figure 15, Figure 14, Figure 16, and Figure 17. The results consistently demonstrate GTA-1B's performance advantages over GQA-1B across all tested configurations.

These findings align with our primary results (e.g NVIDIA A100 80GB GPU), further reinforcing GTA-1B's scalability and adaptability across various hardware platforms. Notably, the I/O-bound decode phase shows significant benefits owing to GTA-1B's optimized memory access patterns. Collectively, these results provide robust evidence for the practical utility of GTA-1B in diverse real-world deployment scenarios.

## C.4 ADDITIONAL PRACTICAL INFERENCE DEPLOYMENTS

In this appendix, we provide detailed information about our experimental setup and present additional benchmark results for GTA-1B, GQA-1B and MLA-1B under half-precision computations.

We conducted comprehensive benchmarks using the `transformers` library (version 4.36.0) to evaluate the practical performance of our models across various hardware platforms. The experimental setup included the following specifications:

- **Hardware:** NVIDIA H100 80GB, NVIDIA A800 80GB, NVIDIA RTX 3060 12GB, Apple M2, and BCM2712
- **Precision:** Both full-precision (FP32, main text) and half-precision (FP16/BF16, this appendix)
- **Input Lengths:** 128, 512, 1024 and 2048 tokens

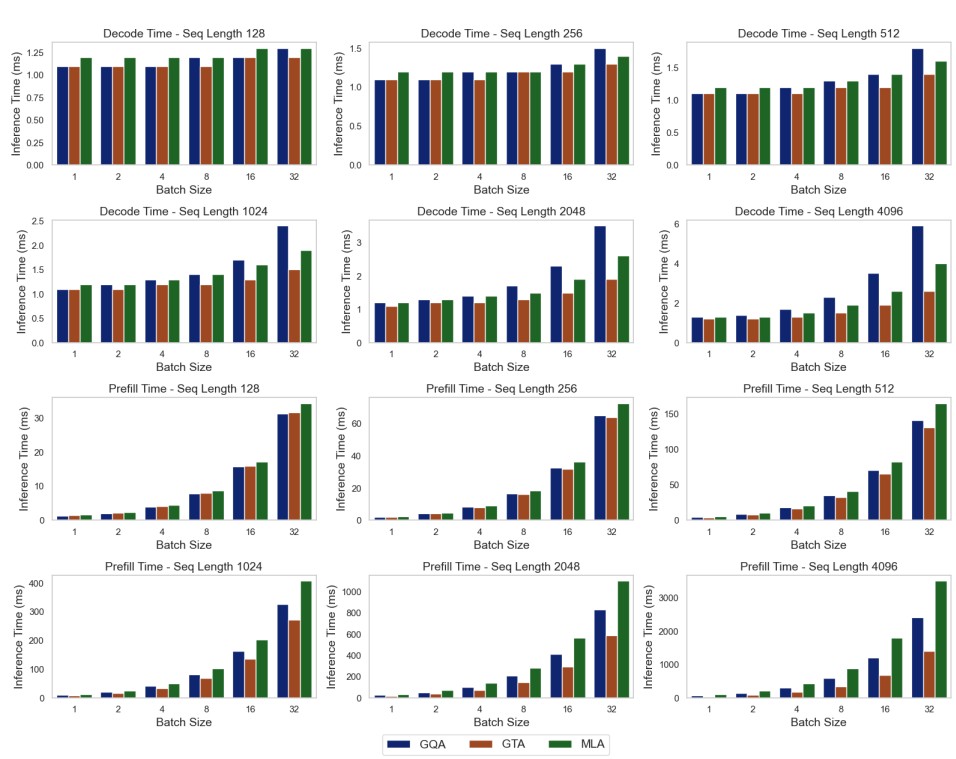

Figure 14: Prefill and decode times across configurations on an NVIDIA A100 80GB GPU.

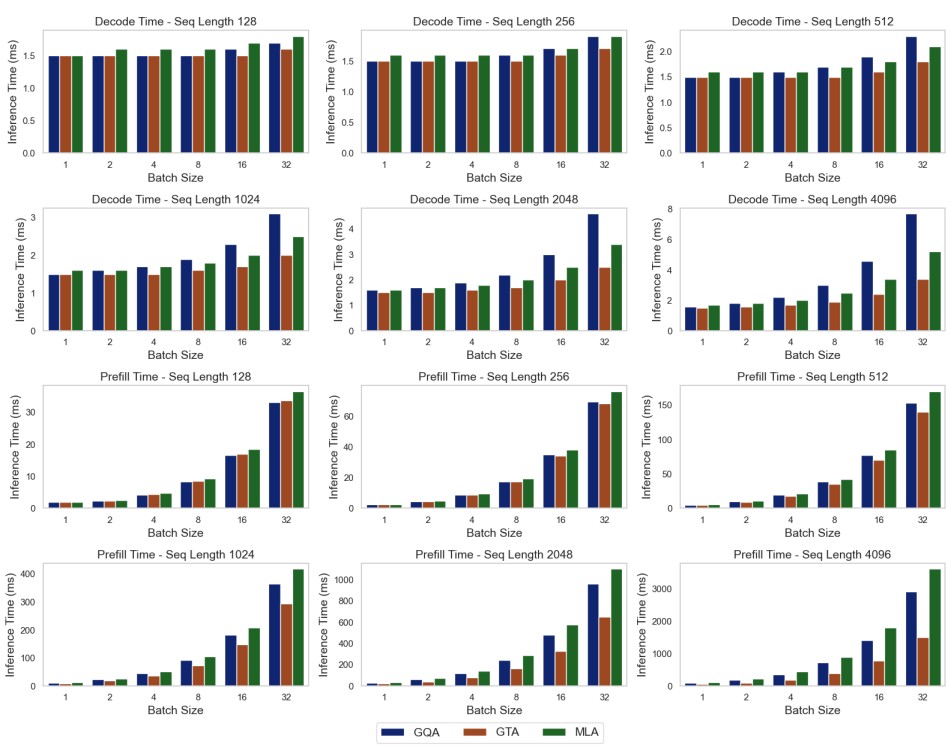

Figure 15: Prefill and decode times across configurations on an NVIDIA A100 40GB GPU.

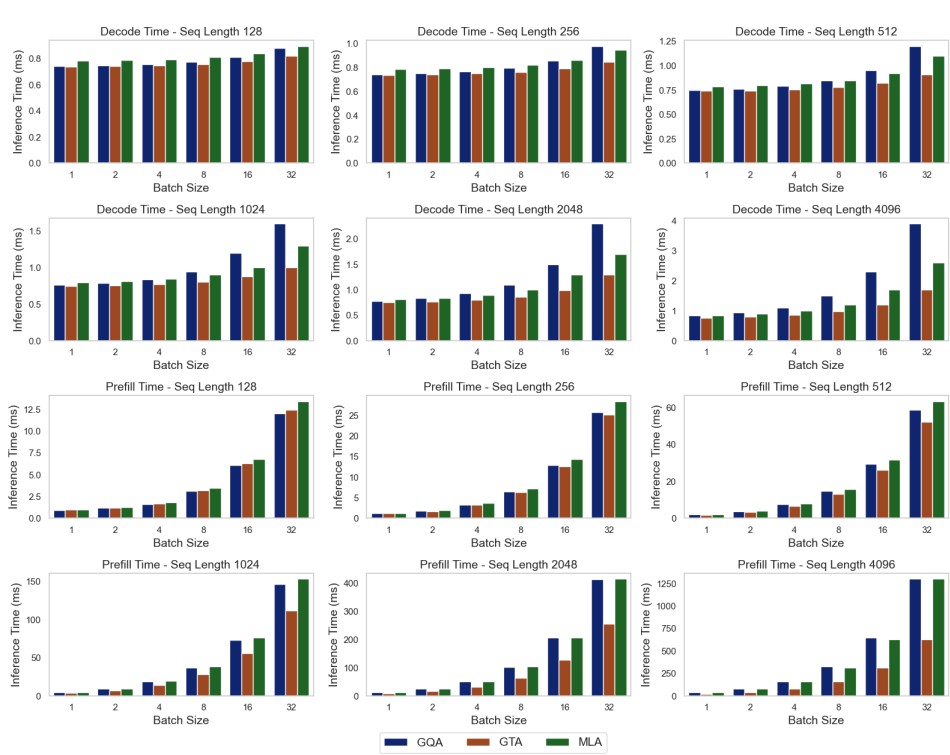

Figure 16: Prefill and decode times across configurations on an NVIDIA H100 80GB GPU.

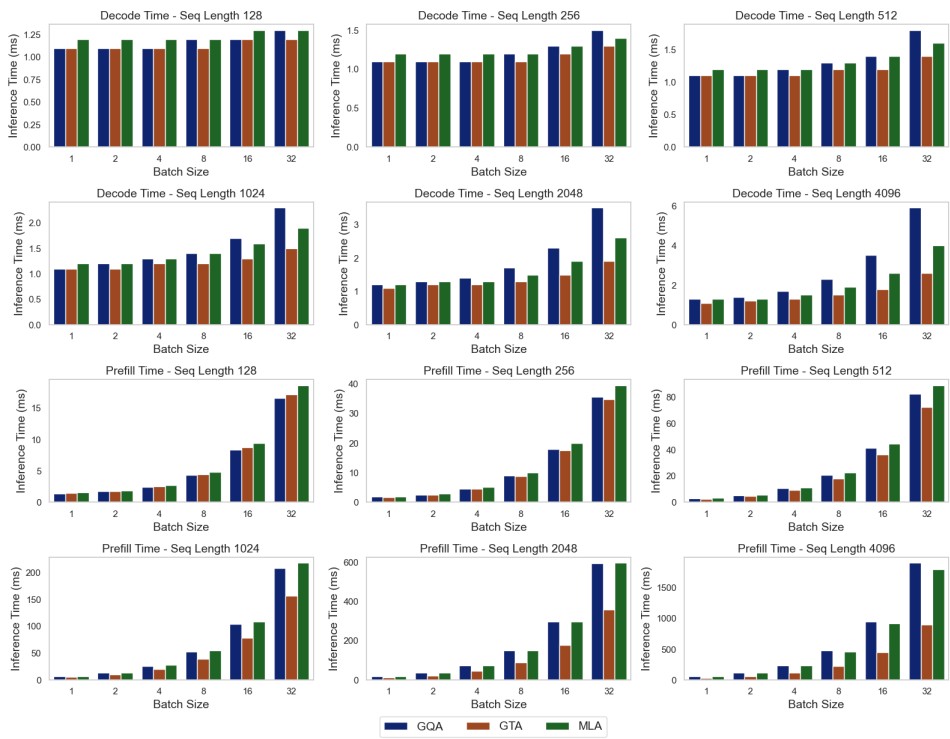

Figure 17: Prefill and decode times across configurations on an NVIDIA H100 PCIe 80GB GPU.

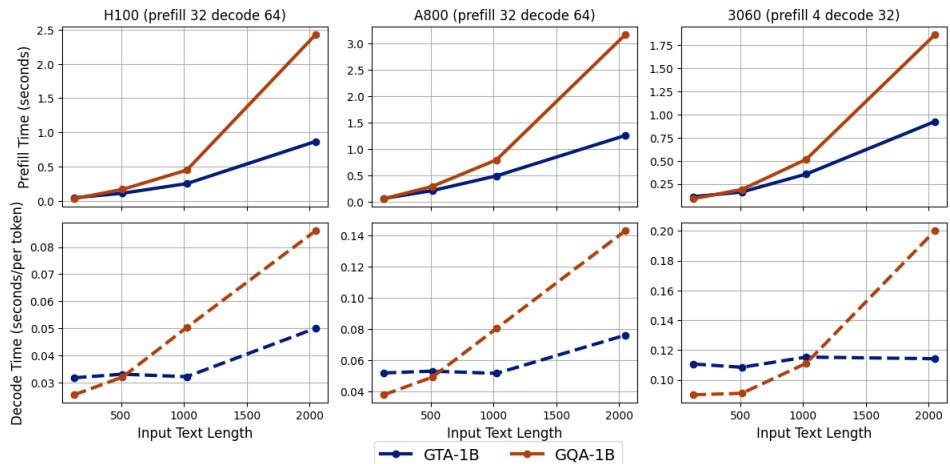

Figure 18: Half-precision prefill and decode times for GTA-1B and GQA-1B across configurations on NVIDIA H100, NVIDIA A800, RTX 3060.

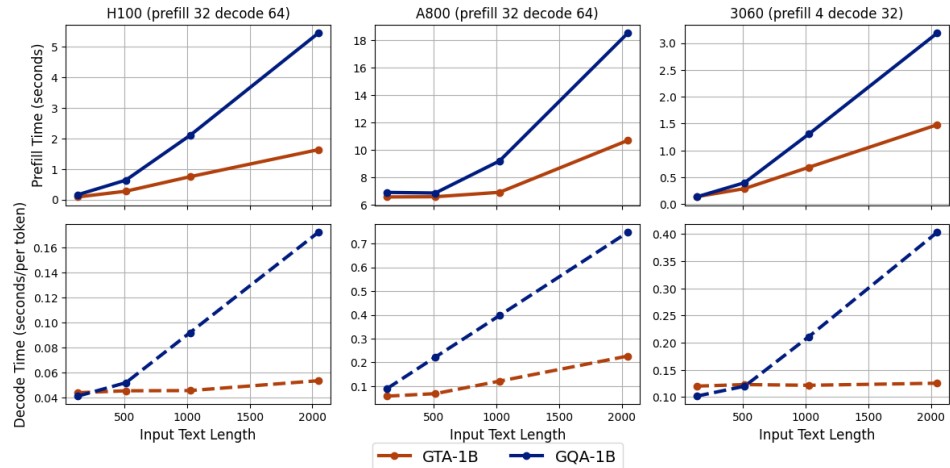

Figure 19: Half-precision prefill and decode performance of GTA-1B and GQA-1B models with cache offload, evaluated on diverse hardware platforms across various test configurations.

For KV cache implementation, we used two approaches:

- **Standard benchmarks:** `DynamicCache` (default in transformers)
- **Offload benchmarks:** `OffloadedStaticCache` (allocates fixed memory, pre-caches two layers on GPU)

All results represent the average of three stable runs after a warm-up phase.

While the main text presented full-precision results, here we provide complementary half-precision benchmarks that demonstrate similar performance patterns but with overall improved efficiency across all hardware platforms.

Figure 18 shows half-precision performance without cache offload. Similar to full-precision results in the main text, GTA-1B (blue solid line) consistently outperforms GQA-1B (orange dashed line). The performance advantage becomes more pronounced at longer sequence lengths, with GTA-1B demonstrating improved efficiency in both prefill and decode phases.

Figure 19 presents the half-precision results with cache offload enabled. GTA-1B's efficiency advantage is further enhanced in this memory-constrained scenario, especially on the NVIDIA A800

80GB at longer sequence lengths. This confirms that GTA-1B's optimized memory access patterns are particularly effective in I/O-bound scenarios, consistent with the full-precision findings reported in the main text. These half-precision benchmarks demonstrate that GTA-1B maintains its performance advantages over GQA-1B across different precision settings, validating the architecture's practical efficiency for real-world deployment scenarios.

## D    THE USE OF LARGE LANGUAGE MODELS

We used large language model (LLM) solely for grammar and spelling checking. The LLM did not generate, refine, or select research ideas, hypotheses, methods, analyses, results, or conclusions, and it did not write substantive content. All scientific contributions, experiment design, data interpretation, and writing decisions are the authors' own. The authors take full responsibility for any remaining errors.

