# OpenReview forum: "Grouped-head latenT Attention"
_ICLR.cc/2026/Conference — ICLR 2026 Conference Withdrawn Submission_

### Official Review · Reviewer_xXpZ · 2025-10-17

**Soundness:** 2
**Presentation:** 3
**Contribution:** 2
**Rating:** 2
**Confidence:** 5

**Summary:**

This paper introduces Grouped-head latent Attention (GTA), an attention mechanism designed to improve the efficiency and scalability of Transformer models. GTA restructures standard multi-head attention by grouping heads into shared latent representations and incorporating a non-linear gating mechanism that adaptively fuses information across groups.

Experiments across several benchmarks—language modeling, summarization, and translation—show that GTA achieves significant speedups (up to 2×) and comparable or better accuracy than GQA, MQA, and standard multi-head attention. The paper claims that the gated latent structure allows GTA to retain the expressive capacity of multiple heads while improving efficiency.

**Strengths:**

1.  The paper is clearly written, with well-organized sections and visual explanations (e.g., Figure 3). The motivation and methodology are easy to follow, which makes the technical contribution accessible.
2. The proposed GTA module demonstrates impressive results across multiple benchmarks. It achieves both higher performance and substantial computational gains, showing it can be a strong replacement for standard attention mechanisms.

**Weaknesses:**

1. The introduced non-linear gating mechanism appears largely orthogonal to the KV cache design and resembles existing gating structures such as GAU (Gated Attention Unit), gate design in Linear Attention variants (RetNet, Mamba and GLA for example) and recent Gated Attention.
Given the large performance improvement shown in Figure 3, it’s unclear whether the gain truly comes from the grouping strategy or simply from the addition of the gate—which is a well-known enhancement that could be applied to GQA or MLA as well. This raises doubts about the novelty of the contribution.

2. The experiments use fixed configurations for GQA and GTA. However, in practice, one could freely adjust the grouping strategy to achieve a desired tradeoff between speed and performance. The paper does not explore this design space or provide analysis of flexible grouping choices. For instance, how much performance degradation occurs under MQA (single-query) settings?

3. Since gated attention mechanisms already exist and can yield similar performance boosts, the paper should clarify what distinguishes GTA conceptually and practically from MQA and GQA. If GTA’s advantage lies mainly in grouping and caching efficiency, the claim that it “improves performance via latent fusion” requires further justification, especially since GTA also achieves 2× speedup in value computation. The performance gain from latent sharing in Figure 3 seems marginal.

**Questions:**

1. How does GTA perform under different grouping configurations (e.g., varying number of latent heads or groups)? Could tuning these parameters yield different tradeoffs between accuracy and efficiency, and are there guidelines for optimal design?
2. In MQA-like settings, how much performance degradation occurs when the number of query heads is reduced? Does GTA maintain robustness under more aggressive compression?
3. Since gated attention mechanisms can already improve efficiency and performance, what is the fundamental advantage or distinctive property that makes GTA superior in principle, beyond empirical results?

---

### Official Review · Reviewer_v7kN · 2025-10-31

**Soundness:** 2
**Presentation:** 3
**Contribution:** 3
**Rating:** 4
**Confidence:** 3

**Summary:**

The paper proposes Grouped-Head Latent Attention (GTA), a new Transformer attention mechanism designed to improve the efficiency of LLMs by exploiting head-wise redundancy in multi-head attention. GTA introduces two key innovations:

* Shared attention maps across heads, allowing multiple heads to reuse the same attention scores, thereby reducing the key cache and computational overhead.
* Nonlinear value decoding, where compressed latent value representations are decoded through a learned, context-adaptive (input-aware) sigmoid gate to generate head-specific values.

**Strengths:**

+ The description of the training setup is detailed and contributes positively to reproducibility.

+ The experiments are comprehensive, covering accuracy and efficiency comparisons across diverse models of different scales and attention mechanisms, as well as evaluations of SFT training performance.

**Weaknesses:**

- This paper asserts that "attention maps across heads display high similarity", which is an important motivation for the grouped-head attention design. However, this claim is not quantitatively supported. The authors do not provide empirical analysis or visualization demonstrating head-level similarity, such as correlation heatmaps, cosine similarity, or variance across heads. Without such evidence, the assumption of head-level redundancy remains speculative and undermines the justification for sharing attention maps across heads.

- Compared with MLA, this work does not appear to demonstrate particularly significant innovation. First, from an algorithmic perspective, the idea of sharing attention scores across heads is not a novel finding. Experimentally, under similar parameter scales and cache sizes, GTA shows only marginal improvement over MLA, as shown in Table 1.

- GTA1, GTA2, GTA3, GTA4 are not clearly described in Table 1 and 2. The authors should include key configurations in the main text.

**Questions:**

* Why does MLA introduce high computational overhead? The matrix absorption in MLA should not incur additional computation.

* How does GHA in Figure 1 illustrate the concept of "sharing attention scores across heads"? The figures of GHA and GTA can be further improved for a quick understanding.

---

### Official Review · Reviewer_u5gL · 2025-11-01

**Soundness:** 3
**Presentation:** 3
**Contribution:** 2
**Rating:** 6
**Confidence:** 4

**Summary:**

This paper proposes Grouped-Head Latent Attention (GTA), an attention mechanism designed to reduce the computational and memory overhead of transformer-based LLMs while maintaining model performance. GTA organizes representations into groups (nq query groups, nk key groups, nc value groups) and uses latent compression (dimension dl ≥ dh) combined with nonlinear gating to maintain expressivity while reducing memory footprint from O(2HN) in MHA to O((nkdh + ncdl)N).

**Strengths:**

1. Design Space: The paper presents a useful design space that explores the trade-offs between efficiency and expressivity (MHA, GVA, GHA, GTA). This framework provides useful intuition that might help guide future research.

2. Models from 160M to 8B parameters are evaluated on sequence lengths from 2K to 128K tokens and multiple benchmarks: 12+ downstream tasks spanning reasoning, coding, and instruction-following are considered. The paper also goes over a detailed complexity analysis with formal derivations, followed by the simulated LLM-Viewer framework for optimal inference modeling and finally, real PyTorch implementations covering both compute-bound (prefill) and I/O-bound (decode) scenarios.

3. Hardware Diversity: GTA is evaluates on 6+ platforms spanning server GPUs (H100, A800, A100), consumer GPU (RTX 3060), and edge devices (Apple M2, BCM2712), with both standard and cache-offload scenarios.

**Weaknesses:**

1. The paper evaluates inference efficiency but ignores training costs. There's no training time comparison between GTA and GQA/MLA and no analysis of convergence speed (do GTA models require more/fewer training steps?). It would also help to analyze memory consumption during training.

2. Long context evals: Models trained on 2K-4K tokens are evaluated on up to 128K tokens (Appendix C.1), but no long-context task performance metrics (e.g., retrieval tasks, needle in haystack) are shared. Figure 12 shows only latency, not accuracy degradation at long contexts. Without long-context training or evaluation on quality, it's hard to confirm that the architecture efficiency gains do not compromise long-context understanding.

3. GTA introduces multiple architectural choices (nq, nk, nc, dl) but provides no principled guidance for setting these. Tables 5-6 show different configurations (GTA1, GTA2, GTA3, GTA4) with varying hyperparameters. There's no systematic study of how to choose these values for new model scales.

**Questions:**

1. Table 3 shows GTA-1B scores 39.56 vs GQA-1B's 40.62 average accuracy (-1.06 points, 2.6% degradation), but after SFT recovers to 42.17 vs 40.64. Why does GTA-1B underperform GQA-1B at base model stage but outperform after SFT? Is this pattern consistent across multiple training runs, or could it be a random seed effect?

2. Should we expect to always need SFT to recover performance, or can the base GTA match base GQA?

3. The paper evaluates inference but omits training costs. What is the wall-clock training time comparison between GTA, GQA, and MLA for the 1B models? What is peak memory usage during training and does GTA require more training steps to reach the same validation loss as GQA?

4. Models trained on 2K-4K tokens are tested at up to 128K tokens (Figure 12, Appendix C.1), but only latency is reported. What is perplexity at different context lengths (4K, 8K, 16K, 32K, 64K, 128K) for GTA vs GQA? How does GTA perform on long-context benchmarks (e.g., SCROLLS, L-Eval, needle in haystack) and does GTA's compression cause quality degradation at longer contexts compared to GQA?

5. Section 3.1 and Figure 1 introduce Grouped-Value Attention (GVA) and Grouped-Head Attention (GHA) as intermediate designs, but they're never evaluated empirically. Can the authors provide empirical results for GVA and GHA as ablations, which would validate their design considerations.

---

### Official Review · Reviewer_UsqW · 2025-11-03

**Soundness:** 3
**Presentation:** 3
**Contribution:** 2
**Rating:** 6
**Confidence:** 4

**Summary:**

The paper proposes an efficient attention method that reduces the memory usage and increase throughput using grouping of query and key projections across heads and a value decoder to compress value cache. This design is shown to reduce the KV cache by 70% while preserving the model quality up to 1B model size in LLMs. This is important especially  for long-context or resource-constrained scenarios. Overall GTA achieves a favorable trade-off between efficiency and expressivity, maintaining strong modeling capacity while reducing resource demands.

**Strengths:**

- Paper explains the different attention architectures well and the difference of the proposed GTA method is described well
- The method shows strong performance, even lower eval loss than multi-head attention despite having less number of attention parameters

**Weaknesses:**

- Novelty of the method us small, is an incremental improvement over GHA method
- The evaluation seems sound however there is no baseline comparison with the GHA method. Ablation section is good but it would have been great to include GHA baseline in Table 1 & 2.
- It is not clear why SFT improves performance of GTA more than GQA, there is no intuition or explanation provided.

**Questions:**

- For Table 1, the configurations of GTA 1 and GTA 2 is not explained. What’s the difference between them?
- It seems after SFT, GTA outperforms GQA, but without SFT GTA falls behind (Table 3). What is the reason SFT improves the performance of GTA more than GQA?

---

### Note · Authors · 2025-12-05

I have read and agree with the venue's withdrawal policy on behalf of myself and my co-authors.